https://doi.org/10.1038/s41467-018-07788-5　　**OPEN**

# Hidden heterogeneity and circadian-controlled cell fate inferred from single cell lineages

Shaon Chakrabarti[1,2,3], Andrew L. Paek[4,9], Jose Reyes[4], Kathleen A. Lasick[5], Galit Lahav 🔘 [4,6,7] & Franziska Michor 🔘 [1,2,3,6,7,8]

The origin of lineage correlations among single cells and the extent of heterogeneity in their intermitotic times (IMT) and apoptosis times (AT) remain incompletely understood. Here we developed single cell lineage-tracking experiments and computational algorithms to uncover correlations and heterogeneity in the IMT and AT of a colon cancer cell line before and during cisplatin treatment. These correlations could not be explained using simple protein production/degradation models. Sister cell fates were similar regardless of whether they divided before or after cisplatin administration and did not arise from proximity-related factors, suggesting fate determination early in a cell's lifetime. Based on these findings, we developed a theoretical model explaining how the observed correlation structure can arise from oscillatory mechanisms underlying cell fate control. Our model recapitulated the data only with very specific oscillation periods that fit measured circadian rhythms, thereby suggesting an important role of the circadian clock in controlling cellular fates.

[1] Department of Biostatistics and Computational Biology, Dana-Farber Cancer Institute, Boston 02215 MA, USA. [2] Department of Biostatistics, Harvard T. H. Chan School of Public Health, Boston 02115 MA, USA. [3] Department of Stem Cell and Regenerative Biology, Harvard University, Cambridge 02138 MA, USA. [4] Department of Systems Biology, Blavatnik Institute, Harvard Medical School, Boston 02115 MA, USA. [5] University of Arizona, Tucson 85721 AZ, USA. [6] Broad Institute of Harvard and MIT, Cambridge 02139 MA, USA. [7] Ludwig Center at Harvard, Boston 02215 MA, USA. [8] Center for Cancer Evolution, Dana-Farber Cancer Institute, Boston 02215 MA, USA. [9]Present address: University of Arizona, Tucson, 85721 AZ, USA. These authors contributed equally: Shaon Chakrabarti, Andrew L. Paek. Correspondence and requests for materials should be addressed to G.L. (email: Galit_Lahav@hms.harvard.edu) or to F.M. (email: michor@jimmy.harvard.edu)

Elucidating the mechanisms of cell cycle control has been one of the most important endeavors in cell biology over the last decades. Since the seminal discoveries of the *cdc* and *wee* genes in yeast and the introduction of the idea of cell cycle checkpoints[1–3], much effort has been devoted to characterizing the genes and proteins that act in concert to regulate the cell cycle[4]. An important breakthrough in this regard has been the recognition that the circadian rhythm likely plays a crucial role in cell cycle control. While historically the cell cycle has been considered to be independent of the circadian clock, there is emerging evidence that these two processes may be intricately connected, with the circadian clock providing an extra layer of control on the cell cycle[5–7]. Not surprisingly, the coupling between the circadian clock, cell cycle and cell death pathways (or the lack thereof) has major implications for anti-cancer therapies[8–10], and forms the basis of the emerging field of cancer chronotherapy[11]. Whether any coupling exists in different cancer types, the possible phenotypic outcomes of such a coupling, and how it can potentially drive heterogeneous cellular responses to cancer therapies remain fundamental questions to be addressed.

A recent study[12] proposed that correlation structures in the inter-mitotic times (IMT) of cells, which have been observed in several experiments over the past decades[12–17], could be generated as a result of circadian gating of the cell cycle. The origin of these intricate correlation structures among cellular lineages has been the subject of intense study, since they are expected to act as key probes into the underlying biochemical and physical processes governing cell cycle dynamics[12–18]. The recently proposed circadian model can in principle capture the observed correlations in IMT, including the widely varying mother–daughter relationships and the so called cousin–mother inequality[12,19] (where the cousin correlation in IMT is greater than the mother–daughter correlation), but it does not account for the distinct shapes of IMT distributions that have consistently been observed in previous studies[20,21]. Inferring these distributions from single cell data is a challenging task in scenarios with multiple possible fates due to biases introduced in the observed data as a result of stochastic competition among cellular fates[22]. Current methods of inferring these distributions do not account for this competition effect[20], and hence are applicable only in limited scenarios where a single fate dominates—for example when drug concentrations are very low or very high. In addition, there is evidence for the existence of strong correlations among times to death of sister and cousin cells[22–26]. However, all previous computational approaches describe mechanisms that specifically explore correlations in either IMT or apoptosis times (AT), and do not provide a unified approach to explain the experimental observations in a comprehensive manner. Existing models therefore cannot explain the entire set of observations obtained from single cell lineage tracking experiments.

Here we set out to design an integrative method to address these fundamental issues. We generated single cell lineage tracking data of human colorectal cancer cells (HCT116), both in the absence and presence of the chemotherapeutic agent cisplatin, to explore lineage correlation structures in IMT and AT of cells. We found complex correlation structures both in IMT and AT, which depend on the degree of relatedness of the cells. Interestingly, we also found that related cells display a large degree of similarity in p53 dynamics and cell fate after cisplatin treatment, providing strong evidence that cellular heterogeneity prior to drug treatment predisposes cells to specific fates. This result is reminiscent of previous work on TRAIL-induced apoptosis[24] and proliferation-quiescence fate choices in cells[27,28], and suggests that heterogeneous levels of proteins passed on from mother to daughter cells can to a large extent determine cell fates early in the daughter cell's lifetime. Based on this result, we developed a theoretical model in which the phase of a cellular oscillator at the time when a mother cell divides controls eventual cell fate probabilities in the daughters. To investigate the ability of this theory to explain our experimental observations, we developed two computational algorithms: (1) a general statistical method to quantify the large extent of drug-induced hidden heterogeneities in IMT, which cannot be directly observed in the data due to stochastic competition between cell division and death events[22], and (2) a computational algorithm to mimic single cell lineage tracking experiments allowing for oscillatory control of cell fates. We showed that this integrative method, using a minimal set of tunable parameters, can explain the entirety of the correlation structures in addition to accounting for hidden heterogeneities. Importantly, using the same theoretical formulation while switching to physically realistic but non-oscillatory models of cell fate control failed to recapitulate the cousin-mother inequality. In addition, our model was not able to reproduce the correlation structures for most values of the oscillation period, except for a period of 24 h and a few other multiples of 12 h such as 12 and 48 h. Our work therefore suggests an important role of the circadian clock in controlling times to cellular fates, both in the presence and absence of drugs, and provides a widely applicable method for correctly inferring heterogeneities in times to cell fate from single cell data.

## Results

**Correlation structures before and after cisplatin treatment.** In order to obtain accurate single cell lineage data on cell fates and times to cell fates, we used HCT116 p53-VKI human colon cancer cells, a previously established clonal cell line in which one allele of the endogenous *TP53* gene is tagged with the Venus fluorescent protein[29]. We imaged untreated, proliferating HCT116 p53-VKI cells for two days, followed by a switch to fresh media with 12.5 μM cisplatin. Time lapse microscopy and lineage tracking was then continued for another three days after cisplatin administration, and times at which cell divisions and death events took place were recorded throughout (Fig. 1a, Supplementary Figure 14 and 15, Supplementary Movie 1). Intermitotic and apoptosis times (IMT and AT, respectively) were defined from the time a cell is born to the time of mitosis or death (Fig. 1b). We classified these events into three categories—events that occur entirely before the time of cisplatin administration ($T_d$), events that straddle $T_d$, and those that occur after $T_d$ (Fig. 1b).

By computing correlation structures in times to division before cisplatin administration (Fig. 1c–e, Supplementary section 1), we found that the mother–daughter correlation in IMT is insignificantly different from 0 (Pearson correlation, $\rho \sim -0.03$ for 71 pairs, $P$-val [$t$-test] = 0.7, 95% CI [$-0.26$, 0.16]), sister correlations are large ($\rho \sim 0.73$ for 80 sister pairs, $P$-val [$t$-test] = $2.9 \times 10^{-14}$, 95% CI [0.6, 0.8]), and the cousin–mother inequality[12,30] (where the cousin correlations are larger than the mother–daughter correlations) is satisfied ($\rho \sim 0.34$ for 46 cousin pairs, $P$-val [$t$-test] = 0.02, 95% CI [0.1, 0.57]). Here sister cells are defined as cells with the same mother while cousins are cells whose mothers were sisters. For division events straddling $T_d$ (red cells in Fig. 1b), we observed similar correlations among sisters and cousins, though smaller in magnitude (Supplementary section 1, Supplementary Figure 1). Note that for these events mother–daughter relationships are not defined, since the mothers are not part of this category. The apoptosis times (AT) of sister and cousin pairs of cells treated with cisplatin (red and green cells that die in Fig. 1b) also show significantly positive correlations (Fig. 1f, g; $\rho \sim 0.64$ for 93 sister pairs, $P$-val [$t$-test] = $3.09 \times 10^{-12}$, 95% CI [0.48, 0.78]; $\rho \sim 0.38$ for 60 cousin pairs, $P$-val [$t$-test] = 0.001, 95% CI [0.15, 0.54]).

We then explored correlations in cell fates after cisplatin administration (Fig. 2a–b). We found that sister cells shared the same fate (death or survival) about 80% of the time, regardless of whether sisters divided before or after cisplatin treatment (Fig. 2a, b). If cell fates were independent, sisters would

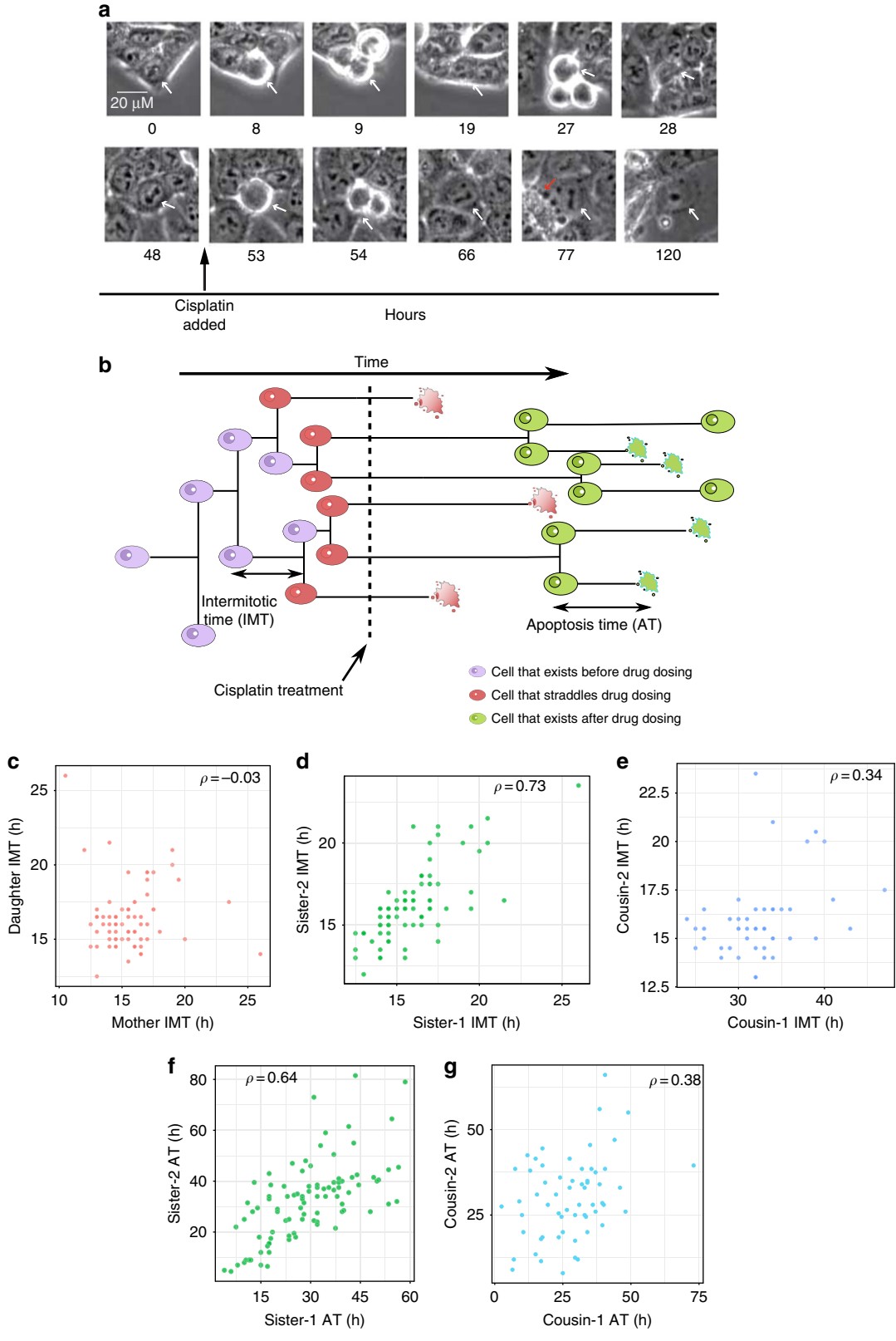

be expected to share the same fate ~53% of the time (see Supplementary section 2 for calculations). The similarity in fates of related cells diminished with increasing numbers of divisions separating the cells (Supplementary Figure 2a). Cells separated by four divisions (3rd cousins) shared the same fate in similar

proportions to unrelated cells. We observed similar trends when cell division following cisplatin treatment was also incorporated into cell fate considerations (Supplementary Figure 2b, c). To rule out possible spatial effects, such as similar cisplatin exposure levels of physically proximal cells driving the similarity in sister

**Fig. 1** Correlations in HCT116 cells before and after cisplatin treatment in a single cell lineage-tracking experiment. **a** Example of live-cell imaging of a single cell before and after cisplatin. The white arrow points to the cell tracked. The red arrow at hour 77 highlights an apoptotic cell. Images are shown for each cell division. Scale in top left image is 20 μm. **b** Cartoon representation of the time-lapse microscopy experiment. Cells that are born and divide before cisplatin addition are colored purple, cells born before cisplatin treatment that eventually divide or die after treatment are red, and cells that exist purely after cisplatin administration are in green. **c–e** Lineage correlations in inter-mitotic times of cells existing before cisplatin treatment (purple cells in **b**). Pearson correlations ($\rho$) are shown on top of each panel, and colors for lineage correlations are maintained throughout the text. The mother–daughter correlation is $\rho = -0.03$ for 71 pairs, $P$-val = 0.7, 95% CI [−0.26, 0.16]. The sister correlation is $\rho = 0.73$ for 80 pairs, $P$-val = $2.9 \times 10^{-14}$, 95% CI [0.6, 0.8]. The cousin correlation is $\rho = 0.34$ for 46 pairs, $P$-val = 0.02, 95% CI [0.1, 0.57]. Cousin correlations are higher than the mother–daughter correlation, a phenomenon called the cousin–mother inequality[12]. **f, g** Lineage correlations in times to death of cells treated with cisplatin (red and green cells in **b**). Note that by definition mother–daughter pairs do not exist for cells that die. $\rho \sim 0.64$ for 93 sister pairs, $P$-val = $3.09 \times 10^{-12}$, 95% CI [0.48, 0.78]; $\rho \sim 0.38$ for 60 cousin pairs, $P$-val = 0.001, 95% CI [0.15, 0.54]. Statiistical significance of the correlations was computed by a $t$-test (Supplementary section 1)

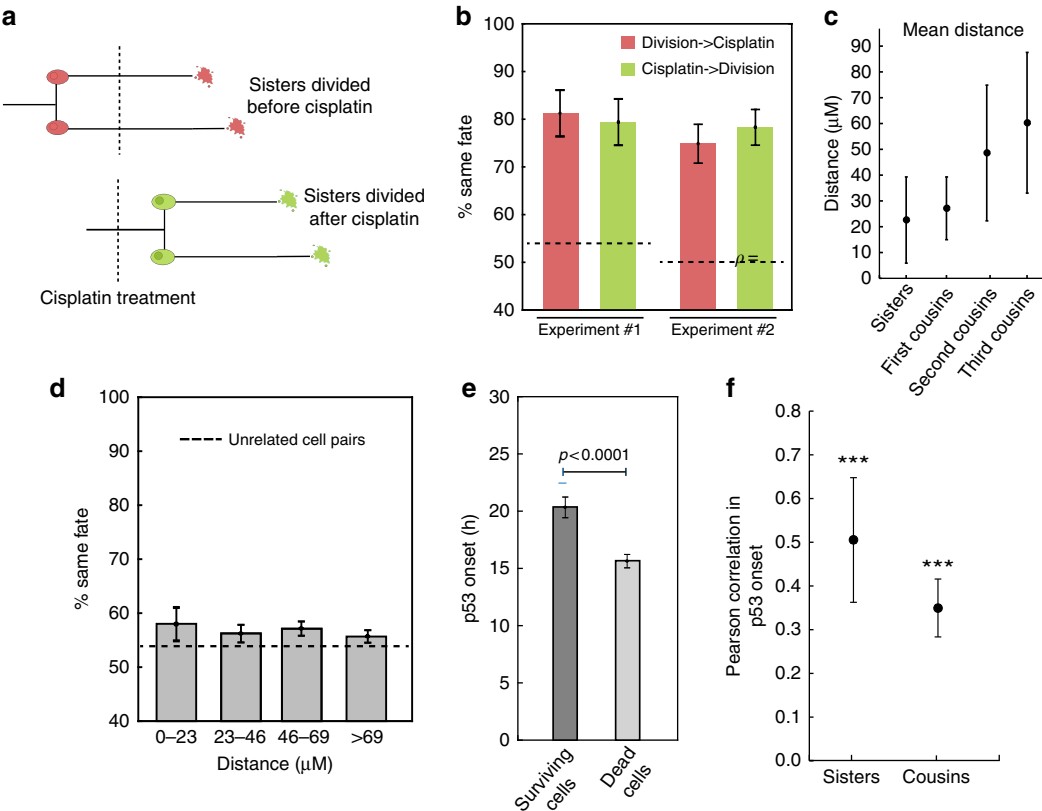

**Fig. 2** Cell fate and p53 dynamics are correlated in sisters and cousins. **a** Sister cell pairs were divided into two groups: those that divided before or after cisplatin treatment. **b** The percentage of sisters in each group that share the same fate. Experiment #1 $N = 61$, $N = 108$, for experiment #2 $N = 150$, $N = 150$. The dashed lines represent the % of unrelated cells that share the same fate. **c** Mean distance separating cells when cisplatin was added by relationship $N = 61, 259, 414, 533$. The centroid of the nucleus was used for the location of each cell. Euclidean distances were computed for every pair of cells. **d** % of unrelated cell pairs that share the same fate grouped by distance separating cells when cisplatin was added. $N = 243, 896, 1341, 1791$. Sister cells were on average 23 μM apart. The dashed line is the same as in **b** Error bars for **c**, **d** are standard deviation. **e** p53 onset in apoptotic cells was faster than in surviving cells. $N = 144, 250$. Error bars represent standard error of the mean. Significance by $t$-test (**f**) p53 onset was correlated among sister and cousin cells. ***$P < .0001$. See methods for calculations of significance

fates, we measured distances between related cells. Though sister cells tend to be close together in space (Fig. 2c), unrelated cells separated by similar distances do not exhibit the same degree of similarity in fates (Fig. 2d). This observation suggests that shared fate is not the result of proximity-related factors but rather cell-intrinsic factors that predispose cells to a particular fate. Using a geminin reporter, we ruled out potential connections between cell cycle stage at the time of cisplatin treatment and cell death (Supplementary section 3, Supplementary Figure 3). However, cells in G1 during cisplatin treatment were more likely to remain arrested following treatment than cells in G2/M

(Supplementary Figure 3i). Finally, we found that p53 dynamics was correlated between related cells (Fig. 2e, f) and was also correlated with the time to death (Supplementary Figure 3h), consistent with our previous work on cisplatin-induced cell fates being associated with p53 dynamics[29]. Taken together, our results suggest that the state of a cell prior to cisplatin exposure, likely inherited from its mother during mitosis, affects the rate of p53 accumulation and predisposes it to a specific cell fate. This finding motivated the development of our birth-death process models and lineage simulations, as discussed below.

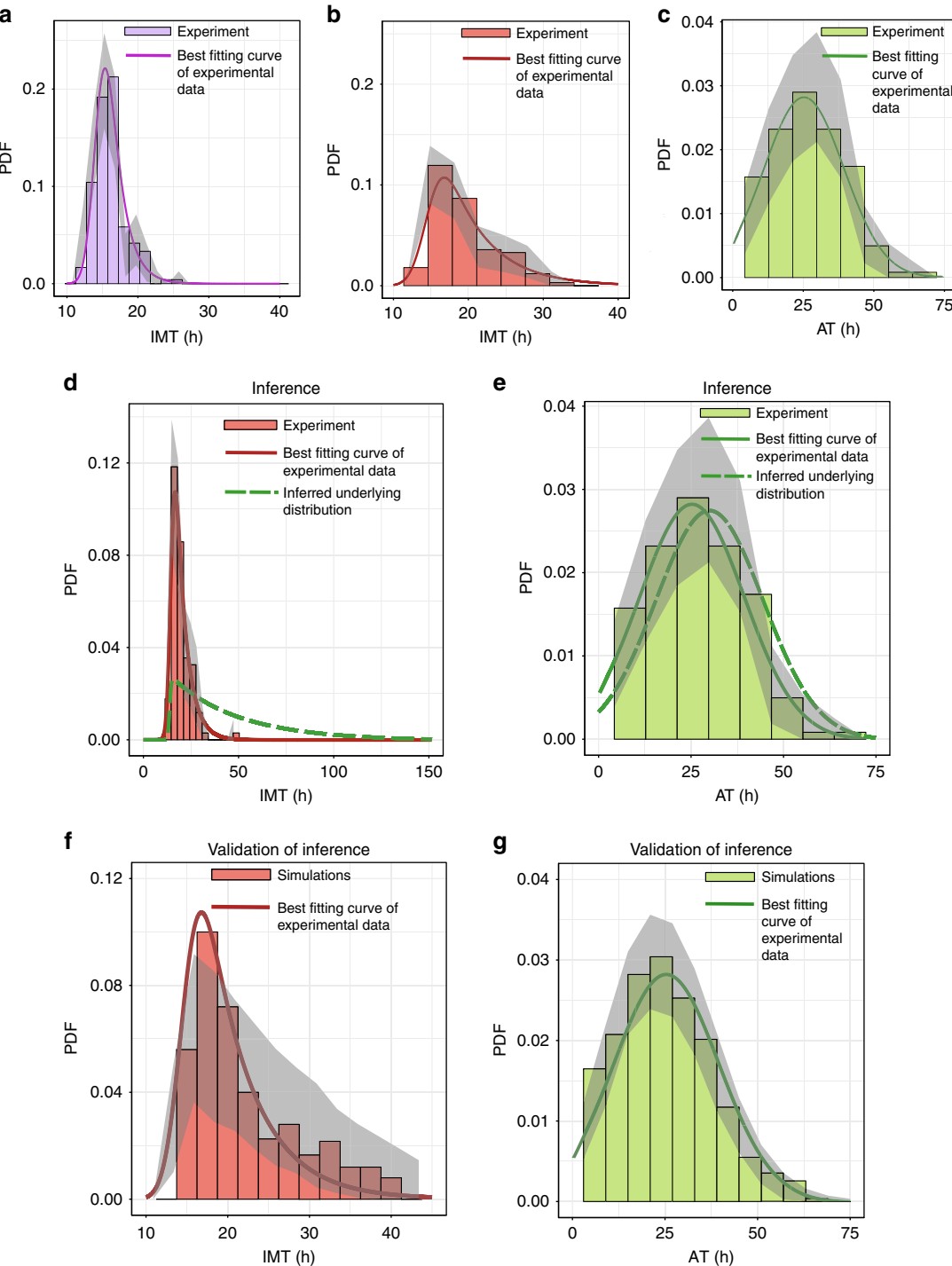

**A statistical algorithm to quantify hidden heterogeneity.** To develop a mechanistic understanding of these lineage correlation structures in HCT116 cells, it is crucial to correctly quantify and account for the large heterogeneities in IMT and AT. However, this is a challenging task in the presence of multiple competing cellular fates[22]. The true underlying distributions governing cell division and death processes are masked due to stochastic competition between the fates, and the observed experimental data (Fig. 3a–c) may therefore be very different compared to the true underlying distributions[22]. The cause of this bias is the mutual exclusivity of cellular fates—the only fate that is observed is the one that happens to occur earlier. Hence values chosen from the right tails of the true IMT and AT distributions are unlikely to be

observed due to the earlier occurrence of the competing fate. As a result, the observed times to both division and death are skewed towards shorter times; the extent of this bias depends on how much the underlying IMT and AT distributions overlap.

In order to infer the correct underlying distributions of IMT and AT, we developed a computational framework to model the times to cell fates in the single cell lineage data, accounting for the large sister correlations. In brief, we described the entire dataset (the single cell data is provided as Supplementary Data 1 with a detailed explanation of the data structure in Supplementary Data 2) as a collection of sister pairs with concordant or discordant fates, and designed a likelihood function to compute the probability of observing the data. The basic form of the

**Fig. 3** Quantifying hidden heterogeneity induced by cisplatin. The color code follows Fig. 1b. **a** Probability density function (PDF) of the IMT before cisplatin treatment, with a mean of 16.1 h. **b** IMT PDF of cells straddling the cisplatin administration event. Mean time is 20 h, indicating a slowing down of the cell cycle after cisplatin administration. As explained in the main text, this is a biased estimate of the mean cell cycle time. **c** Apoptosis time PDF measured directly from the data. The experimental data in **a**–**c** are shown as histograms derived from 160, 104, and 186 data points respectively. The corresponding best-fitting Exponentially Modified Gaussian (EMG) distributions are shown as solid curves. Gray shaded areas represent 95% confidence intervals generated from 1000 bootstrapped samples of the data. Parameters for the curves are given in Supplementary section 4. **d, e** Experimental data and inferences from our algorithm. **d** The inferred IMT distribution after cisplatin addition is shown as a green dashed curve. The inferred heterogeneity using our statistical model (standard deviation of the green dashed curve) is 33.05 h while existing inference techniques[20] using the red histogram would have incorrectly concluded 5.65 h. **e** The inferred apoptosis time distribution after cisplatin is shown as a green dashed curve. As expected for a scenario where the average death rate is higher than the division rate, the inferred time to death distribution is not heavily biased, unlike the inferred IMT distribution in **d**. **f–g** Validation of our inferences using birth-death process simulations. **f** The histogram represents one example of the observed IMT distribution from our birth-death process simulations, using the data generating the green dashed lines from panels **d** and **e** as inputs. The close match between the histogram and the red solid line representing the data validates our inference procedure and inferred IMT distribution. **g** Similar to **f**, but for the apoptosis time distribution. Parameters for the inferred distributions (dashed lines) are given in Supplementary Table 5 and parameters obtained from fits to the data (solid red or green lines) are given in the Supplementary section 4. The gray shaded areas in **f**, **g** denote 95% confidence intervals generated from 500 simulations

likelihood function for one sister pair is given by:

$$f_i(t_1^i, t_2^i; \boldsymbol{\theta}) = c_z(1 - S^i(t_1^i), 1 - S^i(t_2^i))S^i(t_1^i)h(t_1^i; \boldsymbol{\theta})S^i(t_2^i)h(t_2^i; \boldsymbol{\theta}).$$

(1)

Here $f_i(t_1^i, t_2^i)$ is the bivariate joint probability density of observing the first sister cell in the $i$th pair to divide or die at time $t_1^i$ after its birth, and the second cell of that sister pair to divide or die at time $t_2^i$ since birth; $S^i(t_1^i)$ and $S^i(t_2^i)$ are the univariate survival functions of the sisters, denoting their probabilities to survive until times $t_1^i$ and $t_2^i$, respectively; $h(t_1^i; \boldsymbol{\theta})$ and $h(t_2^i; \boldsymbol{\theta})$ are the univariate hazard functions of the sisters, representing their risks of dividing (or dying) at times $t_1^i$ and $t_2^i$, respectively (see Supplementary section 5 for details); and $\boldsymbol{\theta}$ is the vector of parameters to be inferred from the data; it depends on the functional form chosen to represent the variability in IMT and AT. We used the Exponentially Modified Gaussian (EMG) function for this purpose, since this function was found to best describe our observed data (Supplementary section 4, Supplementary Tables 1-3). The EMG has also previously been shown to better explain cell division time variability than other commonly used functions[20,21]. The EMG is a convolution of a Gaussian with parameters $\mu, \sigma$ and an exponential with parameter $\lambda$. Finally, we accounted for the large sister correlations by using a copula $c_z$, which is a function that joins together one-dimensional density functions to form a multivariate density function[31] (Supplementary section 5). We used a Gaussian copula throughout this work, parameterized by the single parameter $z$, which represents the Pearson correlation between sister cells. We modeled stochastic competition among cellular fates using a competing risks framework. The full likelihood function is a product of Supplementary Equation (1) over all sister pairs in the dataset. Further details of the model are provided in Supplementary section 5. We observed that accounting for correlations among sisters led to significant improvements in the estimation of the distribution parameters using a simulation approach as well as direct observation of the pre-cisplatin IMT data (Supplementary section 5, Supplementary Figure 4). We also accounted for the cells that survive until the end of the experiment or 72 h after cisplatin treatment, and allowed for the possibility of a delay between the time of drug administration and the realization of its effect on cell fates[30] (Supplementary section 5). Copulas, while commonly used in finance[32], have rarely been used in biology. Our results highlight the usefulness of this method for modeling correlated data in this and potentially other biological contexts.

Our computational framework was first used to identify the underlying IMT distribution of HCT116 cells in the absence of cisplatin. Since there was very little cell death in this scenario, the inferred IMT distribution should be almost identical to a histogram of the IMT data, which is indeed what we found (parameters of the EMG function in Supplementary section 4 and Supplementary Table 4). In addition, since there are a relatively large number of IMT pairs available in the data (80 pairs), a direct calculation of the Pearson correlation of sisters from the data should also be close to the inferred value. As expected, the inferred sister–sister correlation of 0.71 (Supplementary Table 4; standard error calculated as the square root of the parameter variance, computed from the Hessian matrix) was within error identical to the directly calculated value of 0.73 (Fig. 1d). These results provide a direct validation of our inference procedure. The bivariate density of the sisters is also captured well by the copula framework with inferred univariate EMG margins and the inferred Pearson correlation, as demonstrated in Supplementary Figure 5.

We then inferred the drug-induced distributions of IMT and AT, accounting for the measured sister correlations in IMT and AT using the copula framework (Fig. 3d, e; inferred parameters given in Supplementary Table 5 and Supplementary Table 6). Remarkably, we found that the underlying IMT distribution is very different from the distribution obtained directly by binning the data (Fig. 3d): the directly computed mean of the division times of cells that straddle the dosing event is 20 h as opposed to the inferred mean of 47.22 h (Fig. 3d). Similarly, the standard deviation of the observed histogram is 5.65 h, underestimating the inferred but "hidden" heterogeneity with a standard deviation of 33.05 h (Fig. 3d). Current methods for analyzing this kind of single cell data that treat cell division and death independently[20] would therefore severely underestimate the effects of the drug. The inferred distribution of AT (Fig. 3e) is also shifted, though not as much as the IMT distribution, as expected (see Supplementary Section 5 for a detailed discussion).

To independently confirm these results, we used the inferred IMT and AT distributions from Fig. 3d, e as inputs to a stochastic, age-dependent birth-death process simulation of cellular proliferation[33]. Following single cells over time, we generated stochastic waiting times to division or death of each cell based on the hazard functions corresponding to the input IMT and AT distributions (Supplementary section 6). A hazard function, as outlined in the context of Supplementary Equation (1), represents the risk of a cell dividing or dying at any point in time, given that it has survived until that time. The results of these

simulations provide the post-competition IMT and AT histograms (Fig. 3f, g). We observed a close match between the predicted IMT distribution and experimental results (Fig. 3f), providing a confirmation of our inferences. A similarly close match was obtained for the AT distribution (Fig. 3g). Conversely, if the measured IMT and AT distributions (Fig. 3b, c, respectively) were instead used as inputs to the simulation, the results were not found to match the experimental data (Supplementary Figure 6). This observation arises because the observed data only exhibit the post-competition IMT and AT distributions and do not represent the true underlying distributions that generate the observed data. This finding highlights the importance of using an integrative analysis approach like ours to correctly infer the underlying IMT and AT distributions.

**Protein production/degradation models cannot explain correlations**. With the correct IMT and AT distributions inferred as outlined above, we then explored probable mechanistic origins of the lineage correlations in HCT116 cells. Previous work has suggested that cell-to-cell heterogeneity due to the stochastic production and degradation of proteins can influence cell fates and explain the correlation in IMT and AT in closely related cells[25]. We therefore sought to investigate whether such models would also be able to recapitulate the cousin-mother inequality observed in our data prior to cisplatin treatment (Fig. 1c, e). To compute lineage correlations, we added the additional capability of tracking lineages to our simulation framework using directed graphs (Supplementary section 6). In this framework, each vertex in the graph represents a unique cell and directed edges indicate a mother-daughter relationship. We kept track of the birth time and division time of each cell by assigning attributes to each vertex (Supplementary section 6). It has previously been shown that the level of a protein like CDK2 (or the ratio of the levels of two proteins like Cyclin D1 and p21) inherited by daughter cells at the mother's division determines the chance of cell cycle progression versus quiescence[27,28]. To mimic this phenomenon, we generated stochastic trajectories of one protein (called Protein X) or two independent proteins (Proteins X and Y) within each single cell of our simulated lineage trees (Supplementary section 6). The level of Protein X (or the ratio of X and Y) in the mother cell that is passed on to the daughters sets the hazard function for division in our model (see Supplementary section 6 for details). As the level of Protein X at the time the mother divides increases, the probability of longer division times increases for the two daughter cells. As the level of Protein X decreases in the mother, there is an increased probability of shorter divisions for the daughters. In the case of two proteins controlling cell fate, the daughters are more likely to divide slower or faster depending on the magnitude of the ratio of their levels, X/Y. Within this general framework, we investigated a variety of protein production and degradation rates to mimic the fact that different proteins have varying "memory" levels and lose correlation at different timescales[34] (Supplementary section 6).

We found that none of these models were able to generate higher cousin correlations than mother–daughter correlations. Figure 4 shows the correlations obtained for the Protein X only scenario, while the results for the Protein X and Protein Y models can be found in Supplementary Figure 7. As shown in Fig. 4a, when Protein X levels vary widely over time and lose memory of the initial level rapidly, the cousin correlation is less than the mother–daughter correlation and almost equal to zero (Fig. 4b). On the other hand, when Protein X has strong memory of its original state because of very low production and degradation rates (Fig. 4c), not only is the cousin correlation lower than the mother–daughter correlation, but the latter also becomes very strongly positive (Fig. 4d), which does not recapitulate the near zero mother–daughter correlation observed in our data. Similar

results were found for the two-protein case, as shown in Supplementary Figure 7.

In summary, we found that simple models of stochastic production/degradation of proteins and their inheritance across cellular generations, representing our current understanding of cell cycle control mechanisms, cannot explain our observed correlation structures in the HCT116 cell line.

**Unified theory with circadian gating explains correlations**. The HCT116 cell line was shown to exhibit strong circadian oscillations with a period of 24 h[35], and previous experiments suggested circadian control of both the cell cycle[36–38] and cell death[39,40] pathways. Motivated by these experimental observations and studies linking circadian gating to lineage correlations in IMT[12,30], we developed a novel unified theory for explaining the observed correlation structures in HCT116 cells before and after cisplatin dosing. Since we had previously found that approximately 8% of HCT116 cells died over a period of 72 h in the absence of cisplatin[29], we introduced the added dimension of cell death to our simulations (Fig. 5a) and found that while maintaining the correct IMT distribution (Fig. 5b), the origin of the correlations in the absence of drug cannot be ascribed to stochastic competition of fates alone (Fig. 5c). Next, based on our data suggesting that the cellular state inherited by a cell from its mother plays a major role in the decision of apoptosis versus division (Fig. 2a, b), we devised a form of coupling of the circadian clock to the cell cycle and cell apoptosis pathways: both hazard functions of division and death of any cell are determined by the circadian phase at the time the cell was born from its mother. Mathematically this coupling was achieved by introducing the following general structure for the parameter $\mu$ of the EMG:

$$\mu = \mu_0 + A\sin(\Phi), \qquad (2)$$

where $\Phi$ represents the clock phase at the time a particular cell was born, and $\mu_0$ and $A$ are two free parameters. An example plot of $\mu$ as a function of $\Phi$ and hazard functions of three cells born at different phases of the clock is shown in Fig. 5d (see Supplementary Equation 18 and Supplementary section 6 for further details.). We modeled the circadian clock as a sinusoidal wave of period 24 h, corresponding to a clock phase ranging from 0 to $2\pi$. For cells born between the $\pi$ and $2\pi$ phases of the clock corresponding to the second half of the circadian day, the probability to divide or die at earlier ages is increased (pink dot and line in Fig. 5d represent the risk of division; similar curves describe the risk of death). For cells born during the remainder of the phases (0 to $\pi$–the first half of the day), the probability is decreased (yellow dot and line in Fig. 5d). These probabilities were again modeled using hazard functions (Fig. 5d, Supplementary sections 5 and 6). This method of coupling the circadian clock to the cell cycle and cell death pathways via the hazard function is the defining aspect of our model, since it allows us to maintain the correct IMT and AT distributions as inferred from the data. The branching process model with this added gating mechanism was able to quantitatively reproduce the lineage correlations and the cousin–mother inequality observed in the pre-cisplatin part of the experiment (within 95% confidence intervals, Fig. 5e, f). Crucially, this model also reproduces the experimentally observed IMT distribution (Fig. 5g) and requires just one free parameter to recapitulate the correlation structures in addition to the three parameters required to characterize the IMT distribution (Supplementary section 6). Note that our results are robust to small phase differences between mother and daughters at the time of division[36,41] (Supplementary section 6, Supplementary Figure 8). Furthermore, our model does not require the circadian clock of all cells to be synchronized (Supplementary section 6, Supplementary Figure 9).

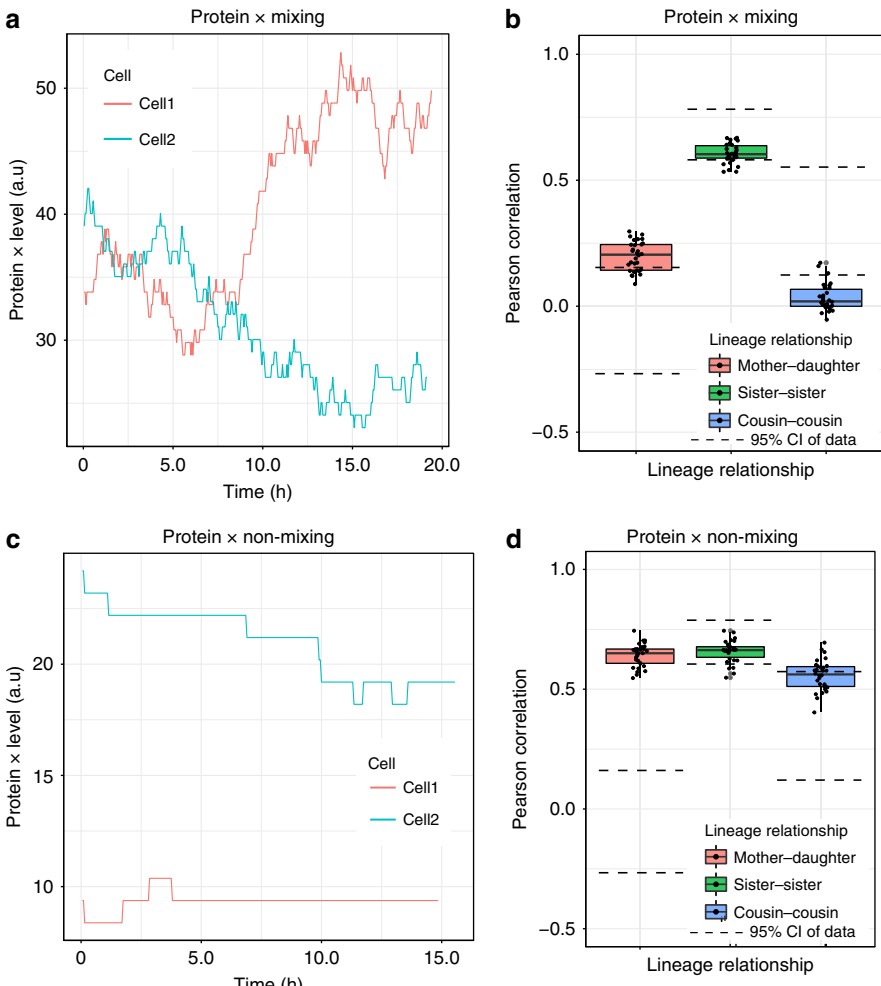

**Fig. 4** A simple model of cell division control by fluctuating protein levels cannot recapitulate the cousin-mother inequality. **a** Levels of protein X as a function of time during the lifetimes of two cells. The protein is said to be "mixing" since the production and degradation rates are high, leading to loss of memory of the initial protein level over the cellular lifetime. **b** Lineage correlations from 30 simulations (shown as black dots) generated by a model where the Protein X level passed on from mother to daughter cells control the hazard function for division of the daughters. As can be seen, the cousin-mother inequality cannot be recapitulated by this model and the mixing property of Protein X leads to negligible cousin correlations. **c** Protein X levels as functions of time in two cells when the protein is "non-mixing": in this case, the production and degradation rates are low and hence the memory of the initial protein level is retained at the end of the cell's lifetime. **d** Lineage correlations from 30 simulations (black dots) for the case of non-mixing Protein X. Once again the cousin–mother inequality cannot be explained, and the non-mixing property of Protein X leads to very large mother–daughter correlations. Parameters for the models in both **b** and **d** were chosen to recapitulate the correct sister correlation as observed in our experimental data (details in Supplementary section 6). The boxplots represent the 1st, 2nd, and 3rd quartiles of the lineage correlations generated in the simulations

The clock-driven correlations, as described above, were obtained by assuming a period of 24 h for the oscillations that couple to the cell cycle. We next investigated whether our model would be able to reproduce the correlation structure with other oscillation periods, since oscillatory processes distinct from the circadian clock have also been suggested to affect cellular proliferation[42]. To this end, we varied the oscillation period in our simulations, choosing the tunable parameters in a way that reproduced the sister correlations and IMT distribution observed in the data (Supplementary section 6; parameters are provided in Supplementary Table 7). Interestingly, we found that only certain multiples of ~12 h time-periods (approximately 12, 24, 48 h; not 36) could reproduce the experimentally observed correlation structure. For all other periods tested (for example 3.5, 6, 14, and 18.5 h), either one of two problems arose: (1) the mother–daughter correlation became strongly positive, for example with 14 and 18.5 h periods (Fig. 6 a, b and Supplementary Figure 10) or (2) at very small time periods like

3.5 h, the cousin correlations reduced to almost zero (Fig. 6c, d). An intuitive explanation for these observations is provided in Fig. 6 a, c. The mother–daughter correlation is set by the interplay between the variable cell cycle lengths and the period of oscillations of the clock. The HCT116 cell line has an approximately 16 h average cell division time. As shown in Fig. 6a, this cell cycle time along with an 18.5 h oscillation period would be expected to generate a strongly positive mother–daughter correlation as daughters are born in a similar part of the circadian cycle as their mothers. On the other hand, when the oscillator frequency is high (time period ~ 3.5 h), the heterogeneity in the cell division times will result in cousins being born at randomly different phases of the oscillator, thereby leading to negligible cousin correlations (Fig. 6c). These intuitive expectations are backed up by our simulation results which incorporate the correct heterogeneity in cell division times, and hence suggest that the circadian clock with a 24 h time period is likely to have generated the observed correlation structure.

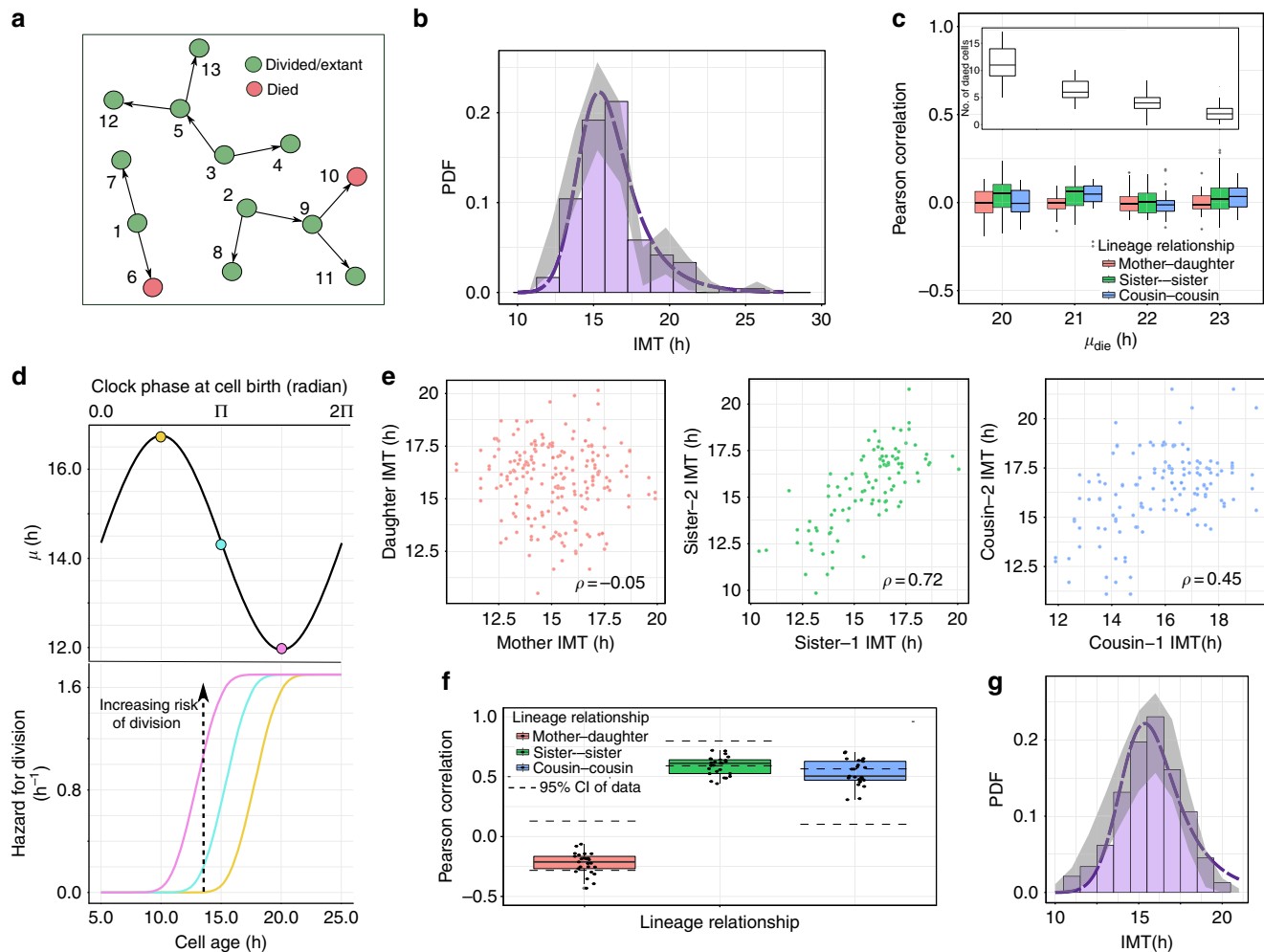

**Fig. 5** Coupling of the cell cycle to the circadian rhythm is required to explain correlations in the absence of cisplatin. **a** Birth-death process simulations keeping track of lineage relationships. Three ancestor cells and their progeny are shown here as examples. Directed edges represent mother–daughter relationships. **b**, **c** Results of a null model with no circadian gating. **b** The IMT distribution before cisplatin (purple dashed line, EMG parameters in Supplementary Table 4) is almost identical to the experimental data (histogram, EMG parameters in Supplementary section 4). The gray shaded area represents 95% confidence intervals generated from 1000 bootstrap samples of the data. **c** Pearson correlations between cell pairs as $\mu_{die}$ of the EMG function is varied. The inset shows the number of dead cells in 25 simulation runs, for different values of $\mu_{die}$. **d–g** Model incorporating circadian gating of the cell cycle and results. **d** A model for the coupling of the circadian clock to the cell cycle. As the phase of the clock at cell birth varies between 0 and $2\pi$, the parameter $\mu$ of the EMG function for division varies, thereby controlling the hazard for cell division (top). Three hazard functions corresponding to three phases of the clock are shown in matched colors (bottom), modeling different division risks for three cells born at different phases of the clock. **e** Instances of Pearson correlations in IMT ($\rho$) generated from the model with circadian gating of the cell cycle. **f** The cousin–mother inequality and the magnitude of all lineage correlations are recapitulated with the model for circadian gating of the cell cycle. The dashed lines indicate the 95% confidence intervals of the IMT correlations as measured from the data. **g** The histogram represents one example of the IMT distribution generated by our simulations incorporating circadian gating. Gray shaded area represents 95% confidence intervals generated from 500 simulation runs. The dashed line represents the inferred IMT distribution, as in **b**. All parameters used for simulated results in **e–g** are provided in Supplementary section 6. All boxplots represent the 1st, 2nd, and 3rd quartiles of the lineage correlations generated from 25 simulation runs

Ultradian oscillations, which have typical periods of ~2–4 h[42] and are common in mammalian cells, are therefore unlikely driving the observed correlations.

We then investigated whether the circadian model could also explain the data arising from cisplatin-treated cells, where there is a significant amount of cell death and the drug induces a large extent of heterogeneity in the cell division and death times (Fig. 3d, e). We used the inferred IMT and AT distributions from our statistical model (Fig. 3d, e; parameters in Supplementary Table 5) as inputs to the birth-death process simulations. Similar to the pre-cisplatin scenario, we first investigated a null model with no coupling to the circadian clock (Fig. 7a, c) and subsequently studied a model with coupling of only the cell cycle

to the circadian clock. We found that while the latter model was capable of generating high correlations in division times, it was not able to explain the experimentally observed magnitudes of correlations among the apoptosis times of either sister or cousin pairs (Supplementary Figure 11). This observation is interesting especially in light of previous work suggesting that correlated IMT of sisters can induce correlations among times to discordant fates of sister cells[22,43]. Our analysis predicts that even if the IMT of sisters are correlated due to circadian gating of the cell cycle, stochastic competition of fates alone cannot induce sufficiently large correlations in apoptosis times of sister or cousin cells.

A model with the circadian clock coupled to both cell cycle and cell death pathways, however, was able to recapitulate the high

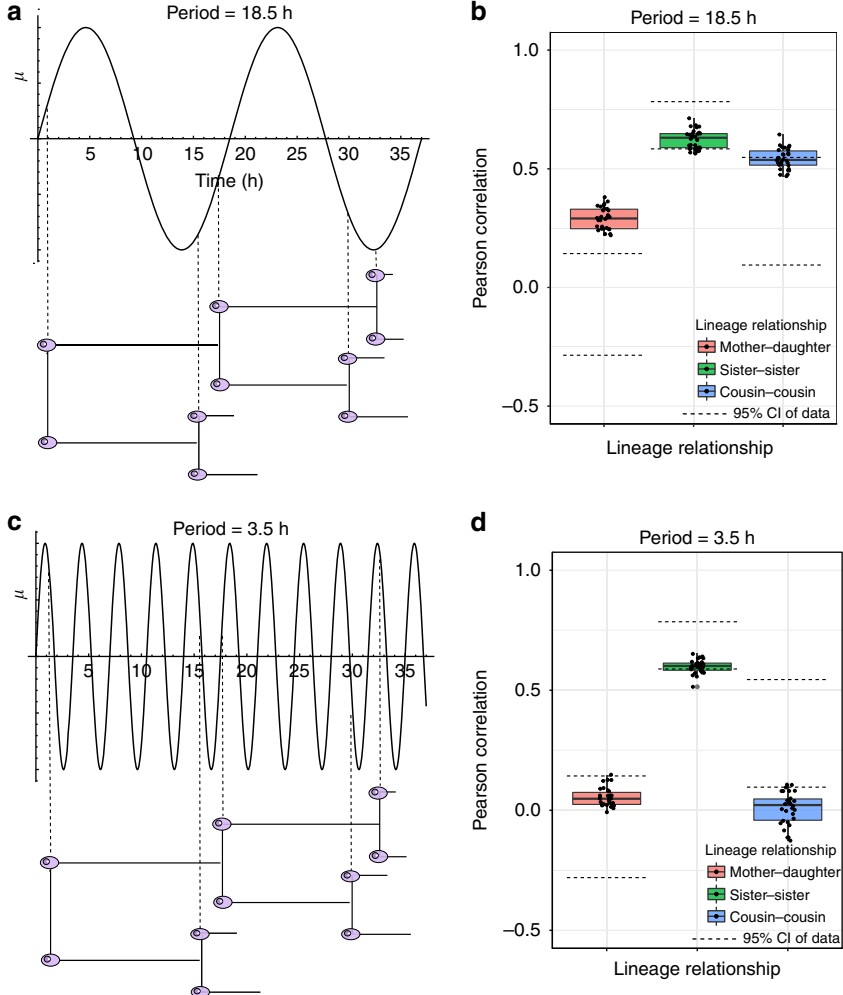

**Fig. 6** Most oscillator time periods fail to capture the correlation structure in intermitotic times. **a** Schematic of an oscillator with an 18.5 h gating of the cell cycle. The inter-mitotic times shown are ~16 h, as observed in the pre-cisplatin HCT116 cells. The standard deviation was chosen as ~2 h to mimic the inferred width of the IMT distribution in Fig. 5b. The dashed vertical lines indicate the phase of the oscillator when a particular cell was born. Similar phases at cell birth result in positively correlated cell division times. This schematic provides an intuitive explanation for the correlation structure obtained from simulations shown in **b**. **b** Lineage correlations obtained from simulations when the oscillator has a time period of 18.5 h. The mother–daughter correlations are larger than observed in the data. **c** Similar to **a** but for an oscillator with a 3.5 h period. The rapid oscillations and IMT heterogeneity combine to result in random phases at which cousins are born, leading to negligible cousin-correlations as seen in **d**. **d** Similar to **b** but for an oscillator with a 3.5 h period. The cousin correlation is negligible and hence does not recapitulate the experimental data. Parameters used to generate these plots were chosen to generate observed sister correlations (details in Supplementary section 6). All boxplots represent the 1st, 2nd, and 3rd quartiles of the lineage correlations generated from 30 simulation runs

correlations in both IMT and AT (Fig. 7b, d). Note that our model predicts that the sister and cousin correlations in IMT after cisplatin addition are smaller than their values pre-cisplatin (Fig. 5f vs. 7b), similar to observations made from the data (Supplementary Figure 1 versus Fig. 1d, e). This decrease in correlations is due to the increased heterogeneity in IMT induced by cisplatin and highlights the importance of accounting for the correct level of variability in cell division (or apoptosis) times. A mechanistic model of lineage correlations therefore must be able to simultaneously account for these heterogeneities. Indeed, our circadian-gating model explains not only the correlation structures (Fig. 7b, d) but also the entire post-competition IMT and AT distributions (Supplementary Figure 12). Finally, to determine if our computational framework can also recapitulate the correlated fates of sisters (Fig. 2b), we introduced correlated random numbers to decide cell fate (Supplementary section 6). We found that this final computational model correctly predicts similarities in sister cell fates (Supplementary Figure 13).

Since the above results suggest that gating of both cell cycle and apoptosis pathways are simultaneously required, we explored the consequences of introducing a phase difference ($\Delta\varphi$) in the gating of the two pathways. When a daughter cell is born, the phase of the circadian clock at that moment could cause both the risks of eventual division or death to increase or decrease in sync (in phase, $\Delta\varphi = 0$). We found that this scenario was able to recapitulate the observed correlations in Fig. 7b, d. However, the circadian phase at the time of cell birth could also in principle increase the chance of division while decreasing that of death, and vice versa (out of phase, $\Delta\varphi = \pi$). A representation of the risks of division and death for these two extreme scenarios, $\Delta\varphi = 0$ and $\Delta\varphi = \pi$, are shown in Fig. 7e. We found that, as $\Delta\varphi$ tends towards $\pi$ (completely out of phase), the IMT correlations between sisters decrease to 0 or even negative values (Fig. 7f), thereby not recapitulating the observed data. These results suggest that the cell cycle and cell death pathways must be gated approximately in phase in the HCT116 cells treated with cisplatin.

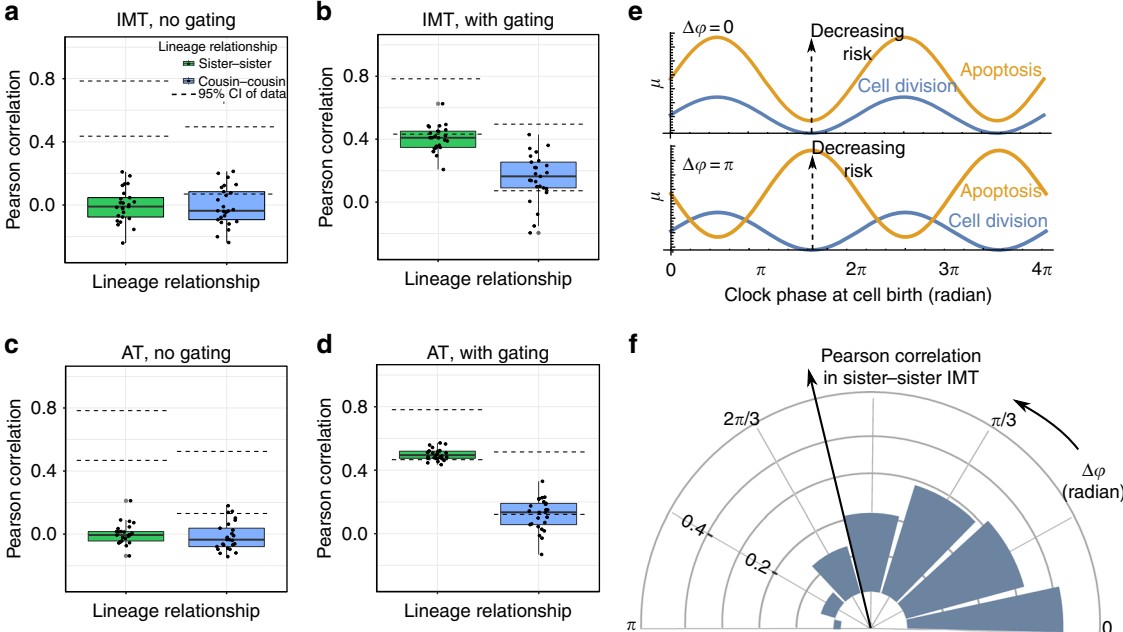

**Fig. 7** In-phase gating of cell cycle and apoptosis pathways by the circadian clock. **a** Analysis of correlations in IMT generated by the model with no circadian gating and **b** with circadian gating of both the cell cycle and apoptosis pathways. **c** Analysis of correlations in apoptosis times (AT) generated by the model with no circadian gating and **d** with circadian gating of both the cell cycle and apoptosis pathways. Note that the high correlations observed in **d** cannot be obtained with a model that has only circadian gating of the cell cycle, and no coupling to cell death (see Supplementary Figure 11). In **a–d** the dashed lines represent the 95% confidence intervals of the respective correlations as calculated from the data while the boxplots represent the 1st, 2nd, and 3rd quartiles of the lineage correlations generated from 25 simulation runs. The fraction of simulation runs resulting in correlation values within the 95% confidence intervals remains approximately the same when increasing the number of simulations from 25 to 150 in **b** and **d**. **e** Two extreme examples of completely in phase gating (top) and completely out of phase gating (bottom) of the cell cycle and apoptosis pathways. The curves represent the parameter $\mu$ of the EMG function. Blue represents $\mu$ for cell division while yellow is the corresponding $\mu$ for cell death. Note that increasing $\mu$ corresponds to decreasing risk, for both division and death. **f** Effect of increasing the phase difference $\Delta\varphi$ between gating of cell cycle and cell death pathways on the sister correlations in IMT. The concentric rings indicate correlation levels and the blue bars denote the median correlations in IMT generated from 25 simulation runs, for different values of $\Delta\varphi$. Parameters used to generate the simulated results are given in Supplementary section 6

Overall, our unified theory provides a comprehensive explanation of the correlation structures in intermitotic and apoptosis times, suggesting that the circadian phase passed on from mother to daughter cells during mitosis controls the chance of eventual division or death of the daughter.

## Discussion

Distributions of cell division and apoptosis times along with lineage correlations among cells are signatures of underlying cellular processes, and deciphering their origins would provide a deep understanding of the fundamental principles of cell cycle control in both normal and cancer cells. However, comprehensive theories that account for both signatures do not currently exist. In addition, correct estimates of the extent of heterogeneity in cell division and death times, which are central to the understanding of both signatures, are difficult to obtain directly from experiments. Existing techniques for inferring these heterogeneities do not account for strong biases that arise from competition among various cellular fates.

To overcome these limitations, we developed a live cell imaging system to track cellular lineages both before and after drug dosing. Using this system, we uncovered complex correlation structures in the times to fate among closely related lineages both before and after chemotherapy administration. In addition, we found that correlations in cell fates and p53 dynamics in response to cisplatin treatment were highest in sister cells, decreasing with time and number of divisions separating cells, suggesting that a cellular state inherited by daughter cells at the time of their

mother's mitosis determines their likelihood of apoptosis. Although in different contexts, conceptually similar conclusions have previously been drawn in the case of TRAIL-induced apoptosis[24] and in the cellular decision of proliferation versus quiescence[27,28]. Together, these results suggest the presence of a heritable cellular state, which determines the probabilities of eventual fate outcomes at the beginning of a cell's lifetime.

We formalized these findings by creating a computational modeling framework in which the heritable cellular state is under circadian control. This approach provides a unified explanation to the entire set of correlation structures in cell division and apoptosis times both before and after treatment. Importantly, we showed that the magnitude of these correlations depends on the extent of heterogeneity in cell division and apoptosis times, and provide a new method to correctly infer their distributions. Our method, unlike previous approaches that treat cell division and death independently[20], can be applied to any cell line treated with arbitrary drug concentrations. There is significant experimental evidence suggesting the control of both cell cycle and apoptosis pathways by the circadian clock[36–40], thereby providing support to our modeling approach. Indeed, we demonstrated that our model was able to recapitulate the correlation structures of intermitotic times only for a few oscillation periods that are approximately multiples of 12 h, including 24 h (Fig. 6 and Supplementary Figure 10). Ultradian oscillations with typical periods of ~2–4 h[42] are commonly observed in mammalian cells and are unlikely to drive the observed correlations. We cannot rule out some of the higher multiples of 12 h as potential time-periods of the oscillations, likely due to the 95% confidence

intervals on the measured correlations being fairly large in our data.

Our circadian gating model is not mutually exclusive from previous protein production/degradation models that have been proposed specifically to explain correlations of sisters and cousins in drug-induced apoptosis times[24,26]. Indeed, since the circadian phase is likely to be passed on from mother to daughter cells via the fluctuating levels of proteins, our model conceptually encompasses the previously proposed mechanisms of generating correlations. While our theory in its current form does not explicitly model circadian protein concentrations and their oscillations, this information is implicitly incorporated in the varying circadian phase of our approach. However, we demonstrate that stochastic protein production and degradation alone cannot give rise to the entirety of the correlation structures, in particular the cousin-mother inequality in IMT (Fig. 4 and Supplementary Figure 7). Since by definition a cousin-mother inequality does not exist for AT, we cannot currently rule out the possibility that correlations in AT of related cells are due to stochastic production/degradation of non-oscillatory proteins. Finally, an advantage of our circadian gating model over previous work[12,30] is that it relies on the birth-death process representation of cellular proliferation. A natural description of asexual reproduction where a single cell divides into two or dies after a stochastic waiting period, the birth–death process is widely used in contexts as diverse as the somatic evolution of cancer, bacterial dynamics, and genome evolution[44–46]. This model allows us to not only recapitulate the correlations observed in single cell data, but also the correct shapes of the highly variable division and death time distributions both before and after drug dosing. Our model is the first to explain such a diverse array of single cell results within one unified framework. It may, however, be possible to envision other more complex models that could explain all lineage correlations without invoking oscillatory mechanisms.

Our model generates exciting predictions for future experimental validation: we predict that both the IMT and AT should become less variable and less correlated if clock proteins are stabilized. This prediction may be directly tested by the addition of drugs like KL001, which have been shown to maintain the CRY protein at high levels throughout a circadian period[47]. In a recent study, deleting the clock genes in cyanobacteria led to a narrowing of the IMT distribution[30], suggesting an important role of circadian gating in establishing the variation in cell division times. It would also be interesting to test our predictions in embryonic stem (ES) cells, which have been shown to develop the circadian clock only in later stages of differentiation as they lose their pluripotency[48]. Comparing correlation structures between ES cells in early versus late stages of differentiation could provide important insights into the consequences of circadian gating on cellular fates.

## Methods

**Cell culture and cell lines**. HCT116 cells were obtained from ATCC and grown in McCoy's with 10% FBS, 100 µg/ml penicillin, 0.25 µg/ml streptomycin and 85 µg/ml Amphotericin. For lineage tracking we used a previously established HCT116 p53-VKI clonal cell line where one allele of the *TP53* gene is tagged at the endogenous locus. A lentiviral H2B-ECFP reporter was used to track cells over time.

**Live-cell microscopy**. To obtain cell lineages we plated approximately 5000 HCT116 p53-VKI H2B-CFP cells to poly-D-lysine coated glass bottom dishes with No. 1.5 thickness (MatTek corporation, P35GC-1.5–10-C) in McCoy's media with 10% FBS. Cells were incubated at 37 °C and 5% CO$_2$ for 72 h to allow cells to attach to the dishes. We then replaced the media with RPMI media lacking phenol red and riboflavin (imaging media) to reduce background fluorescence. Cells were imaged for 50 h in unstressed conditions to establish cell lineages. After 50 h, the media was replaced with imaging media containing 12.5 µM cisplatin and imaged for an additional 72 h to measure p53 dynamics and cell fate. Live cell microscopy was performed in a Nikon Eclipse Ti-E microscope in an enclosure to keep cells at 37 °C, 5% CO$_2$ and maintain humidity. Images were captured using MetaMorph

software every 30 min. We used the following filter sets: Venus—500/20, 515, 520 nM (excitation, beam splitter, emission filter); ECFP—436/20, 455, 480/40 nm. All filters were obtained from Chroma.

**Data analysis**. For cell tracking and image analysis, we used custom made software for Matlab (MathWorks) that allows the user to manually track cell lineages over time using both the H2B-ECFP nuclear marker and Phase contrast images to ensure faithful tracking[49]. Cell identities that were ambiguous were discarded to ensure the reliability of cell lineages. The p53-Venus traces were extracted from background subtracted images in Matlab and are the average of 9 pixels in the center of each nuclei. Cell death and division was identified by morphology in the phase channel, apoptotic cells detach from the glass while the membrane blebs out. Morphology of the localization of the H2B marker also allows identification of cell death and division, with cell death shown by a loss of the normal round nuclear shape and division shown by condensation of the chromatin to mitotic chromosomes and then separation to two separate nuclei. To determine whether cell fate was correlated in related cells a $\chi^2$ test was used to compare the expected and observed portion of cells that share the same fate. For calculations of significance for $R$-values in Fig. 2 and Supplemental Fig. 2, we compared measured $R$-values of related cells to the distribution of 10,000 $R$-values measured from randomly paired cells of equal size.

**Mathematical models and simulations**. The mathematical models and computational algorithms were written in R, version 3.4.0, and Wolfram Mathematica, version 11.0. These are described in detail in the Supplementary Information.

**Code availability**. All code used for the computational modeling will be made available on request to the corresponding authors. The code for lineage tracking and quantifying single cell data can be downloaded at: github.com/balvahal/p53CinemaManual.

## Data availability

The lineage data generated in this study and used for all the analysis is available in Supplementary Data 1 along with an explanation of the data structure in Supplementary Data 2.

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

## Acknowledgements

We would like to thank Mukund Thattai, Jacob Stewart-Ornstein and Bill Forrester for interesting discussions and comments. We also thank the Nikon Imaging Center at Harvard Medical School for help with microscopy. This research was supported by National Institute of Health grants GM083303 and GM116864, and the Ludwig Center at Harvard. A.L.P. was supported by an American Cancer Society New England Division—Ellison Foundation Postdoctoral Fellowship. J.R. received support from CONACyT/Fundacion Mexico en Harvard and a Harvard Graduate Merit Fellowship. The authors gratefully acknowledge support of the Dana-Farber Cancer Institute Physical Sciences Oncology Center (NCI grant U54CA193461).

## Author contributions

S.C., A.L.P., G.L., and F.M. designed and organized the study. A.L.P. performed the experiments. S.C., A.L.P., J.R., and K.L. analyzed the data. S.C. wrote and implemented all algorithms and mathematical models. S.C., A.L.P., and K.L. created figures. S.C., A.L.P. G.L., and F.M. wrote the manuscript.

## Additional information

**Competing interests:** The authors declare no competing interests.

