## [Peer Review File · Nature Communications]

Reviewers' comments:

Reviewer #1 (Remarks to the Author):

The work of Chakrabati et al. follows the fate of cells before and after cisplatin treatment. They first measure the correlations in the inter mitotic times between pairs of cells along lineages. In agreement with previous results, they measure high correlations between sisters and cousins whereas the correlation between mothers and daughters is non significant. They then proceed to analyze the correlations in cell fate (either apoptosis or mitosis) in a similar way. They find that sisters and cousins cells are highly correlated in their fates. They analyze the data in the framework of a statistical method to infer the underlying distributions of the competing cell fates. They develop a simulation based on circadian control of IMT and cell fate that reproduces the data.

The work is very interesting, analyzing cell fate and lineage correlations with and without anti-cancer treatment. It put the idea of chronotherapy into a quantitative framework and suggests different avenues for future work. It should be of wide interest for researchers interested in cell biology, anti-cancer treatments and systems biology. However, the description of the inference and its validation is unclear and should be written more clearly to enable the evaluation of the results. Major and minor comments below:

1. The cell fate takes into account only two outcomes. However a substantial number of cells do not undergo apoptosis nor mitosis. Why not include this third cell fate in the analysis of correlations instead of assuming two fates? Were these cells taken out of the analyses? Also, was there a dependence of that cell fate on the cell cycle phase at cisplatin addition?
2. The statistical analysis enables to take into account the correlations that exist in the lineages and correct them to infer the underlying distribution. In order to verify the validity of the assumptions used for that analysis, an experimental test would be the measure the IMT distribution taking one cell only per lineage. Was this measurement done?
3. The authors discuss in length in the supplementary information why the assumption of equal circadian phase for all lineages is valid. If they want to keep this assumption, they should show that the gating mechanism is apparent in the distribution of cell division events versus time for untreated cells. If cells are synchronized in the clock, the gating should be apparent with time intervals with reduced probability of mitosis events.

4. If the distribution of cell division events versus time does not reflect gating, it may be that this assumption is in fact not needed. It seems that the relevant parameter is the relative phase of the cells compared to the division events. In other words, if the simulation is run with and without this assumption will the results differ? If not (as the authors suggest in the text), why keep this assumption?

5. The evidence for the effect of the circadian clock in this experiment is not strong, as the authors did not interfere with the clock genes nor tried to synchronize the cells (unless evidence is provided in the graph mentioned in comment #3. The circadian clock model does reproduce the results but has many tunable parameters. Therefore, the authors may want to mention this in the discussion.

6. In the figures the time scale should be changed to hours, not frames.

7. Line 198: missing word

8. Line 59: the circadian model was shown to lead to very variable IMT distributions (see Mosheiff et al. PRX 2018), why is it stated here that it does not? Was it meant here that the AT distribution is not?

9. Line 228: missing 24 hours

Reviewer #2 (Remarks to the Author):

In this study the temporal distribution of cell divisions (inter-mitotic times IMT), and apoptosis events (apoptosis times AT) of a strain of colon cancer cells is studied, including the determination of correlations in the timings between sister cells, cousin cells, and between mother and daughter cells. As found in previous studies the authors confirm positive correlations in IMTs and ATs, which they show to be reproduced by a model in which the propensity of cell divisions (and apoptosis) depends on the phase of the Circadian Clock.

The determination of temporal distributions of IMTs is challenging since the measured frequency distribution of IMTs is distorted due to the possibility of cell death, which suppresses the observation of long cell cycle times. To overcome this problem, and to determine the true underlying IMT distributions, the authors perform a thorough statistical analysis based on competing risks and the framework of survival theory and the theory of copulas. This analysis is confirmed by testing a birth-death model with stochastic fate choice (branching process) on the inferred IMT distributions, which reproduces well the data. (the same applies to the analysis of AT distributions)

Since often, although required, such an analysis is not performed, the methodology presented in this study may serve as a standard approach to determine IMT distributions in the future. In particular, the authors demonstrate that neglecting the influence of competing risks can lead to a massive underestimation of the real heterogeneity in cell cycle times. The methodology presented here is therefore a valuable contribution to improve the understanding of cell cycle and apoptosis dynamics in cell populations.

It is intriguing to see that by coupling the cell division and apoptosis timings to the Circadian Clock, the observed sister-sister and cousin-cousin correlations can be reproduced. This hypothesis is indeed very plausible, also given previous experimental evidence for the Circadian Clock influencing cell divisions. This values further investigation, although due to the lack of comparison with other models which may equally well fit these observations, I would refrain from confirming that this hypothesis has been fully validated by this analysis. I assume that the inheritance of protein levels could similarly explain such correlations, and the authors also mention this rightfully in their discussion.

To summarise, this is a very well-written manuscript and the authors succeed in the challenging task (whereby the challenge itself is due to a subtle statistical fallacy) to determine IMT and AT distributions for colon cancer cells in-vitro. Their methodology to use competing risks survival analysis is to my knowledge novel, and should be set as a standard for the analysis of cell cycle times. This alone would render the manuscript worth to publish in Nature Communications. The further modelling of IMT and AT timings correlations, showing that they could be explained by a coupling to the Circadian Clock is a further worthwhile observation, so that I strongly support the publication as a Nature Communications article.

Further minor remarks:

- p.3 line 114: Here it is said that IMT mother-daughter correlations are "weakly negative". However, they are not significantly different from zero, so one cannot say that there are negative correlations.

Furthermore, later in the sentence the "cousin-mother inequality" is mentioned, but not explained what is meant with this.

- Fig. 1, panel (a): It is not visible at which time point cisplatin administration takes place.

- Fig. 2, panel (b+d): No error bars are shown.

- Fig. 2, panel (e): it would be more helpful to show the standard error of the mean instead of the standard deviation of individual measurements. Thereby the error bars would relate more to the shown quantity.

- Many figures, all histograms: No error bars are shown in histograms. Please note that this is necessary if no absolute counts are shown.

Reviewer #1

The work of Chakrabati et al. follows the fate of cells before and after cisplatin treatment. They first measure the correlations in the inter mitotic times between pairs of cells along lineages. In agreement with previous results, they measure high correlations between sisters and cousins whereas the correlation between mothers and daughters is non significant. They then proceed to analyze the correlations in cell fate (either apoptosis or mitosis) in a similar way. They find that sisters and cousins cells are highly correlated in their fates. They analyze the data in the framework of a statistical method to infer the underlying distributions of the competing cell fates. They develop a simulation based on circadian control of IMT and cell fate that reproduces the data.

The work is very interesting, analyzing cell fate and lineage correlations with and without anti-cancer treatment. It put the idea of chronotherapy into a quantitative framework and suggests different avenues for future work. It should be of wide interest for researchers interested in cell biology, anti-cancer treatments and systems biology. However, the description of the inference and its validation is unclear and should be written more clearly to enable the evaluation of the results. Major and minor comments below:

Response: We thank the reviewer for the concise summary of our manuscript and are happy that he/she found it interesting. Our attempt was indeed to build a combined experimental and theoretical framework to understand control principles of cellular proliferation and death of closely related cells, and we are excited that the reviewer believes it will be of interest to a wide group of researchers. We also thank the reviewer for bringing up his/her concerns, each of which we have carefully addressed with new calculations, figures and clarifications. We hope that these changes have improved our manuscript and make it more easily understandable.

1. The cell fate takes into account only two outcomes. However a substantial number of cells do not undergo apoptosis nor mitosis. Why not include this third cell fate in the analysis of correlations instead of assuming two fates? Were these cells taken out of the analyses? Also, was there a dependence of that cell fate on the cell cycle phase at cisplatin addition?

Response: Indeed, a subset of cells (9%) undergoes neither apoptosis nor mitosis following cisplatin treatment. We have now added a table to the supplement, **Fig. S2C**, which shows the percentage of cells for each of 4 observed outcomes after cisplatin: 1. Cells that divided once and survived (27%), 2. cells that died without dividing (28%), 3. cells that divided once and then died (36%), and 4. cells that remained arrested and survived (9%). In the same table we provide evidence that these additional fates are also correlated in related cells. **We discuss these results in a new section in the supplement titled “Cell fate is correlated in related cells”.** In addition, as suggested by the reviewer, we used a Geminin reporter to test whether there is a dependence on cell cycle phase at the time of cisplatin addition and cell fate. We found evidence that cells that are in G1 when cisplatin was added were more likely to remain arrested. For example, although 34% of cells were in G1 at the time of cisplatin treatment, 83% of cells that remain arrested were in G1. We added a table, **Fig. S3i**, that breaks down these

outcomes by cell cycle stage at the time of cisplatin administration and **discuss these results in the main text page 4 and supplemental section 3**. Note that our original conclusion, that cell-cycle stage does not influence cell death, still stands as cells that were in G1 and S/G2 when cisplatin was added died in equal proportions (**Fig S3d**).

2. The statistical analysis enables to take into account the correlations that exist in the lineages and correct them to infer the underlying distribution. In order to verify the validity of the assumptions used for that analysis, an experimental test would be the measure the IMT distribution taking one cell only per lineage. Was this measurement done?

Response: Thank you for this insightful suggestion. In the original version of our manuscript, we had not developed a method for demonstrating the improvement of parameter inference by accounting for correlations. We have now addressed this shortcoming using *two* different methods – (1) using one cell per lineage as suggested by the reviewer, and (2) a simulation approach where we know the true underlying IMT distribution by construction. Both methods demonstrate that accounting for correlations in the data leads to more accurate inference of the underlying distribution. An important point to note is that method (1) underestimates the efficacy of our copula-based approach (for reasons we discuss below) while method (2) highlights the full power of our approach.

Method (1): As suggested by the reviewer, in this approach we assume that the true underlying IMT distribution is represented by a dataset comprising only one cell per lineage, which ensures that there are no correlations in this dataset. Therefore, parameters of the distribution inferred from this uncorrelated dataset will represent the ‘true’ underlying parameters. Correlated datasets of similar size (obtained by taking sister pairs) can then be used to test the usefulness of our copula-based inference approach – our approach used on the correlated data should provide inferred parameters closer to the ‘true’ parameters as compared to inference using the standard method of non-linear least squares (NLS).

By randomly choosing one cell division time from each lineage in the experimental dataset, we obtained 41 inter-mitotic times and inferred the parameters of this IMT distribution (‘true’ parameters). To obtain a correlated dataset of similar size, we randomly chose 20 sister pairs out of the 80 sister pairs in our full dataset. We then used NLS as well as our copula-based approach to infer parameters from the correlated dataset. We compared the similarity of each of the inferred parameters i ($i = \mu, \sigma, \lambda$) to the ‘true’ parameters using the square of a simple distance metric ($D_{NLS,i} = \text{inferred parameter from NLS}_i - \text{true parameter}_i$; $D_{copula,i} = \text{inferred parameter from copula method}_i - \text{true parameter}_i$). Since the 41 independent samples and the 20 sister pairs can all be chosen in many ways, we performed this entire analysis 1000 times, each time drawing a different random set of data and computing the closeness of parameter inferences to the ‘true’ values. We found that our copula-approach consistently outperformed the NLS approach, doing better 64.3% of the time for μ , 56.3% of the time for σ , and 61.1% of the time for λ . These numbers did not change when we increased the sampling from 1000 to 1500 and 2000 times, making it unlikely that the higher frequency of improved parameter estimates occurred by chance.

While the above analysis highlights the improvement achieved by our method, it provides an under-estimate of the improvement that can be achieved. This fact arises because the ‘true’ parameters, inferred from a dataset of just 41 samples, does not perfectly represent what must be the real underlying distribution due to small sample size, and different realizations of drawing 41 independent division times produce slightly different distributions, leading to an increased chance of the NLS method doing better than our copula-based method in specific instances. Therefore, while this method provides a nice way of demonstrating the importance of correlations directly from our data, we investigated a second method in which the true underlying parameter values are unequivocally known.

Method 2: In this method, we used simulations to generate correlated random number pairs (representing sister cells) from an underlying distribution. Hence by construction we know the underlying true distribution. We chose an underlying EMG distribution with the same parameters that described our pre-cisplatin IMT data ($\mu = 28.576$, $\sigma = 2.453$, $\lambda = 0.274$). We then used the NLS method as well as our copula method to infer the distribution parameters from this correlated dataset. As with Method 1, we calculated the distance of parameter estimates from both NLS and copula methods to the true parameter values ($D_{NLS,i}$ and $D_{copula,i}$ respectively) which were used to generate the data. This procedure was again repeated 1000 times. When using 80 simulated sister pairs (to match our experimental numbers), we found that our method was closer to the truth 61.8% of the time for μ , 71.1% of the time for σ and 66.9% of the time for λ . We redid the entire analysis for 100 simulated sister pairs and found that our method performed better 62.4% of the time for μ , 71.4% of the time for σ and 67.3% of the time for λ . In addition, since in this method we have precise knowledge of the true underlying parameters, we also investigated the *magnitude* of improvement in parameter estimation achieved by our copula method over the NLS method. To do this, we defined an improvement metric D_i ($i = \mu, \sigma, \lambda$) given by $D_i = |D_{NLS,i}| - |D_{copula,i}| / \text{true parameter}_i$ for each of the three parameters, whenever $|D_{NLS,i}| > |D_{copula,i}|$. This metric provides a measure of how close the copula inference is to the true parameters, compared to the NLS inference. We found that although there was only $\sim 1\%$ median improvement for μ , the improvement in estimation of the other two parameters was very large: $\sim 14.6\%$ median improvement for σ and $\sim 12\%$ median improvement for λ (see **Fig. S3** for the distributions of D_i).

Thus we have now shown that, using a simulation method as well as an approach directly analyzing the data, accounting for correlations is more likely to produce inferences that are closer to the truth than the commonly used NLS method. **We have added material regarding these results in the main text on page 6 and the detailed analysis in SI section 5 with a new figure, Fig. S3.**

3. The authors discuss in length in the supplementary information why the assumption of equal circadian phase for all lineages is valid. If they want to keep this assumption, they should show that the gating mechanism is apparent in the distribution of cell division events versus time for untreated cells. If cells are synchronized in the clock, the gating should be apparent with time intervals with reduced probability of mitosis events.

Response: Thank you for raising this issue. We apologize that this aspect of our model was not well explained in the original version of our manuscript – we did not mean to suggest that the circadian phases of cells in different lineages are synchronized. In the absence of synchronizing factors like serum shock (which we did not use in our experiments), the phase will almost certainly be poorly synchronized in cells from different lineages. However, since the circadian phase of a mother cell is usually faithfully passed on to the two daughter cells during mitosis, the phase of sister cells *within* a lineage will be very similar at any given time. This observation is the only assumption required in our model to reproduce the lineage correlations. We have now developed new simulations to demonstrate this finding; please see the response to the next question for details. Furthermore, to account for the possibility that the daughter cells may not receive a phase identical to the mother's phase at mitosis due to stochasticity in protein distribution, we also demonstrated in the SI that our model is robust to such small fluctuations.

In our original model, we assumed identical circadian phases across lineages, as pointed out by the reviewer. This assumption was made purely for the sake of simplicity of the model and for reducing computing time and memory requirements. This assumption was not meant to be physically realistic, as we pointed out in the SI, but captured the minimal requirement that sisters have similar circadian phases. Since different lineages are independent in birth-death models such as the one we used, for the purposes of our study it did not influence our results whether or not we assumed identical phases across lineages. This point has now been demonstrated with a modified version of the model, as discussed in response to the reviewer's next question. **We have now added a sentence on Page 8 of the main text to highlight this point, and have added the details of the new simulations to SI section 5.**

4. If the distribution of cell division events versus time does not reflect gating, it may be that this assumption is in fact not needed. It seems that the relevant parameter is the relative phase of the cells compared to the division events. In other words, if the simulation is run with and without this assumption will the results differ? If not (as the authors suggest in the text), why keep this assumption?

Response: Thank you for this observation – indeed, the assumption of identical circadian phase across lineages is not required. As we mentioned in response to the previous question, we used this assumption in our original model only for the sake of simplicity of the model, since making this assumption was a less complicated computational way to maintain the minimal requirement of similar circadian phases of sister cells within a lineage. To demonstrate the validity of our original results on the lineage correlations (in particular the cousin-mother inequality) in the absence of this assumption, we developed a modified model: we chose the circadian phase of the 30 starting cells randomly from a uniform distribution between $[0, 2\pi]$. As a result, the phases of cells across lineages are no longer synchronized at any later time (**Fig. S8a**). We then showed that our original results for the lineage correlations still hold in this modified version of the model (**Fig. S8b**), proving that, as long as the sister cells have similar circadian phases, the experimentally observed correlation structures can be quantitatively

reproduced by our model. Note that in this modified model, the cells *within* a lineage still have synchronized circadian clocks, but we showed in the SI that adding small amounts of randomness in the passing of the phase from mother to daughter did not change our results. **We have now added a sentence on Page 8 of the main text and the details of these simulations to SI section 5.**

5. The evidence for the effect of the circadian clock in this experiment is not strong, as the authors did not interfere with the clock genes nor tried to synchronize the cells (unless evidence is provided in the graph mentioned in comment #3. The circadian clock model does reproduce the results but has many tunable parameters. Therefore, the authors may want to mention this in the discussion.

Response: Thank you for this comment. We indeed did not perturb the clock genes in this study since as previously discussed with our editor, this would have required a significant experimental effort, which is beyond the scope of this manuscript and the timeline for revision. To address the reviewer comment we have now performed additional calculations to show that our model can recapitulate the cousin-mother inequality in the lineage correlations *only* when the period of oscillations is approximately 24 or 12 hours. This estimate is consistent with literature estimates of 24 hours for the clock period in these cells and therefore contributes to validating our results. Additionally, we have developed new models based on the physically plausible mechanism of inheritance of protein levels that could possibly generate these correlations. However, we found that none of these new models were able to reproduce the cousin-mother inequality observed. These additional calculations provide increased support to our claim of the possible role of the circadian clock in controlling cell fate. **These new models are now described and discussed in the revised version of the main text, page 6, and in SI section 5.**

We also thank the reviewer for bringing up the point regarding the number of tunable parameters in our model. We believe that our model has the minimum set of parameters needed to explain the correlation structures, given all the data that our model recapitulates. For example, in the pre-cisplatin scenario, we have only 4 tunable parameters in the model – $\mu_0, A, \sigma, \lambda$ (discussed in detail in the SI). The data that we compare our model predictions with is the IMT distribution as well as the mother-daughter, sister-sister and cousin-cousin correlations. The EMG function for the IMT distribution already requires three parameters for full quantification. Therefore, with the addition of just one more parameter, we can also explain the entire correlation structure. We believe it is quite remarkable that we were able to recapitulate the entirety of the data with such few tunable parameters. **This point is now discussed in the main text, page 8, and in SI section 5.**

6. In the figures the time scale should be changed to hours, not frames.

Response: We have now converted all figure axes to hours. Please note that due to this rescaling of the time by a factor of 1/2, the Y-axis of all figures that display probability densities were rescaled by a factor of 2.

7. Line 198: missing word

Response: Thank you; we corrected this omission.

8. Line 59: the circadian model was shown to lead to very variable IMT distributions (see Mosheiff et al. PRX 2018), why is it stated here that it does not? Was it meant here that the AT distribution is not?

Response: Thank you for this comment; we would like to apologize for the confusing statement. In neither work (Mosheiff et al nor Sandler et al) were the full IMT distributions as carefully studied as they were here. In those papers, the heterogeneity generated by the model was characterized and compared to data using only the coefficient of variation. This is what we meant by the statement the reviewer is referring to. We have now rewritten the relevant line to read: *"The recently proposed circadian model^{12,18} can in principle capture all observed correlations, including the widely varying mother-daughter relationships and the so called cousin-mother inequality^{12,18} (where the cousin correlation is greater than the mother-daughter correlation), but it does not account for the distinct shapes of IMT distributions that have consistently been observed in previous studies^{19,20}".*

9. Line 228: missing 24 hours

Response: Thank you; we have corrected this omission.

Reviewer #2

In this study the temporal distribution of cell divisions (inter-mitotic times IMT), and apoptosis events (apoptosis times AT) of a strain of colon cancer cells is studied, including the determination of correlations in the timings between sister cells, cousin cells, and between mother and daughter cells. As found in previous studies the authors confirm positive correlations in IMTs and ATs, which they show to be reproduced by a model in which the propensity of cell divisions (and apoptosis) depends on the phase of the Circadian Clock.

The determination of temporal distributions of IMTs is challenging since the measured frequency distribution of IMTs is distorted due to the possibility of cell death, which suppresses the observation of long cell cycle times. To overcome this problem, and to determine the true underlying IMT distributions, the authors perform a thorough statistical analysis based on competing risks and the framework of survival theory and the theory of copulas. This analysis is confirmed by testing a birth-death model with stochastic fate choice (branching process) on the inferred IMT distributions, which reproduces well the data. (the same applies to the analysis of AT distributions).

Since often, although required, such an analysis is not performed, the methodology presented in this study may serve as a standard approach to determine IMT distributions in the future. In particular, the authors demonstrate that neglecting the influence of competing risks can lead to

a massive underestimation of the real heterogeneity in cell cycle times. The methodology presented here is therefore a valuable contribution to improve the understanding of cell cycle and apoptosis dynamics in cell populations.

Response: We thank the reviewer for these very positive comments, and we hope that our work will indeed set a standard for analyzing cell cycle distributions in the presence of competing fates.

It is intriguing to see that by coupling the cell division and apoptosis timings to the Circadian Clock, the observed sister-sister and cousin-cousin correlations can be reproduced. This hypothesis is indeed very plausible, also given previous experimental evidence for the Circadian Clock influencing cell divisions. This values further investigation, although due to the lack of comparison with other models which may equally well fit these observations, I would refrain from confirming that this hypothesis has been fully validated by this analysis. I assume that the inheritance of protein levels could similarly explain such correlations, and the authors also mention this rightfully in their discussion.

Response: Thank you for the excellent suggestion of making comparisons with other models. We have now addressed this issue by developing a number of new models based on the stochastic inheritance of protein levels of daughter cells from their mother. Using these new models, we show that our observation of larger cousin correlations compared to mother-daughter correlations (termed the “cousin-mother inequality”) is the crucial aspect of the data that requires invoking an oscillatory mechanism. While the sister and cousin correlations can be explained using simple inheritance rules (as we mentioned in our original manuscript), the cousin-mother inequality cannot be explained with such models.

Additionally, we have now performed new calculations to show that the correlation structures in the data can be recapitulated *only* when the time period of oscillations in our model is set to approximately 24 or 12 hours. This further strengthens our claim that the circadian clock plays an important role in cell fate control, leading to the surprising correlations.

We provide details of the new models below:

(1) *Cell fate probability is affected by one protein:* We first investigated the effect of one stochastically produced and degraded protein on the correlation structure of intermitotic times (IMT). Since the identity of such a protein is not yet fully understood, we called it ‘Protein X’. Previous studies have suggested that this could be one of the many cyclin-dependent kinases, which at high levels lead to faster cell cycle times, or a protein like p21, which at high levels leads to slower cell cycle times (Spencer et al, Cell 2013). Based on these findings, we developed a model in which, in addition to the lineage-generating simulations we had already established (details in the SI), we added stochastic production and degradation of Protein X to each single cell in the simulation. The levels of Protein X in each single cell vary independently and at the time of division, are passed on to the two daughter cells. The level of Protein X that is passed on from mother to the two daughters controls the hazard function for division of both

newborn daughters. As the level of Protein X increases at the time the mother divides, the parameter μ (i.e. the mean of the Gaussian part of the Exponentially Modified Gaussian function) for the two daughters' hazard functions is enhanced, thereby increasing the probability of longer division times for the two daughter cells. If the level of Protein X decreases in the mother, there is increased probability of shorter divisions for the daughters. This behavior is mathematically achieved by making μ a continuous function of the Protein X level. We simulated two sub-models under this Protein X scenario – (a) one in which production/degradation rates of Protein X are high, such that memory of the protein level at cell birth is lost when the cell eventually divides (such proteins are said to be “mixing”; **Fig. 4 a**), and (b) one in which production/degradation rates are very low, such that there are very few production/degradation events in the lifespan of a single cell and hence the memory of the Protein X level that a cell is born with usually remains until the end of that cell's lifetime (**Fig. 4 c**). We chose the parameters of these models such that the sister-sister correlations matched the experimental results. We found that neither model was able to replicate the entire correlation structure observed in the data – the cousin correlations were always lower than the mother-daughter correlations (**Fig. 4 b,d**). Additionally, even though we were able to obtain high sister and cousin correlations in sub-model (b), the strong memory of Protein X caused the mother-daughter correlations to become very high (**Fig. 4 d**), which was inconsistent with our experimental observations.

(2) Cell fate probability is affected by the ratio of two proteins: A recent study demonstrated that the ratio of the level of two proteins (Cyclin D1 and p21) inherited by the daughter cells from their mother determines whether the daughters enter G1 or quiescence with high probability (Yang et al, Nature 2017). We therefore extended the Protein X model previously described to account for the stochastic production and degradation of two proteins, X and Y, in each single cell. The two proteins are independent, and their ratio at the time of cell division determines whether the hazard function of the daughters increases or decreases. Therefore, when the ratio of X to Y is larger than 1, the daughters are more likely to divide more slowly. Under this scenario of two proteins controlling cell fate, we investigated a number of sub-models based on the mixing properties of Protein X and Protein Y: (a) both Protein X and Y are mixing, (b) both Protein X and Y are non-mixing and (c) only one of the proteins is mixing while the other is non-mixing. Under no scenario were we able to replicate the correct correlation structure as observed in our experiments. As with the Protein X only model, the cousin correlations were always less than the mother-daughter correlations and high cousin correlations forced the mother-daughter correlations to become high as well (**Fig. S6 a,b,c**) However, as we demonstrate in the main text, our circadian gating model of the cell cycle can recapitulate the entire spectrum of these lineage correlations in IMT.

In summary, in response to the reviewer's comment, we have developed a number of new models motivated by our current understanding of how inherited protein levels might control cell fate probabilities. We have shown that none of these non-oscillatory models can explain the cousin-mother inequality that we observe in our experiments. We have also shown that the data can be explained only when the time period of oscillations is approximately 24 or 12

hours. **These new models are now described in the main text page 6 and in SI section 5 along with new figures, Fig. 4 and Fig. S6.**

To summarise, this is a very well-written manuscript and the authors succeed in the challenging task (whereby the challenge itself is due to a subtle statistical fallacy) to determine IMT and AT distributions for colon cancer cells in-vitro. Their methodology to use competing risks survival analysis is to my knowledge novel, and should be set as a standard for the analysis of cell cycle times. This alone would render the manuscript worth to publish in Nature Communications. The further modelling of IMT and AT timings correlations, showing that they could be explained by a coupling to the Circadian Clock is a further worthwhile observation, so that I strongly support the publication as a Nature Communications article.

Response: We thank the reviewer for the very positive review of our work, and are excited that he/she believes that this should serve as a standard for future work in understanding cell cycle and cell death times.

Further minor remarks:

- p.3 line 114: Here it is said that IMT mother-daughter correlations are "weakly negative". However, they are not significantly different from zero, so one cannot say that there are negative correlations. Furthermore, later in the sentence the "cousin-mother inequality" is mentioned, but not explained what is meant with this.

Response: We thank the reviewer for pointing out this omission on our part. We have now corrected the statement to read: "...we found that the mother-daughter correlation in IMT is insignificantly different from 0". We have also added an explanation of what the "cousin-mother inequality" means.

- Fig. 1, panel (a): It is not visible at which time point cisplatin administration takes place.

Response: We have now added an arrow in Figure 1A to indicate when cisplatin was added.

- Fig. 2, panel (b+d): No error bars are shown.

Response: We added error bars to figures 2b and 2d. The error bars were created by randomly pairing all cells for each experiment and measuring the % of random pairs that share the same fate. This was repeated 10,000 times and the standard deviation of the averages was used as the error bar. We noted this in the legend of Figure 2

- Fig. 2, panel (e): it would be more helpful to show the standard error of the mean instead of the standard deviation of individual measurements. Thereby the error bars would relate more to the shown quantity.

Response: We have changed the error bars to the standard error of the mean.

- Many figures, all histograms: No error bars are shown in histograms. Please note that this is necessary if no absolute counts are shown.

Response: Thank you very much for pointing out this omission on our part. We have now added 95% confidence intervals to all histograms shown in the main text figures. For the experimental data we obtained the confidence intervals by bootstrapping, while for the simulations we generated the intervals from multiple runs of the simulations. As a result **we have now completely replotted Fig. 3, updated Fig. 5 b and g, and added explanations to the Figure legends.**

Reviewer #3

Chakrabarti and co-workers use single cell lineage data from a colon cancer line to investigate temporal correlations in the generated genealogies and competing risks with a statistical model. They couple the competing risks framework to copula distributions to account for dependences exceeding the correlations induced by the competing risks. Observing that a significant amount of correlation between sister cells cannot be explained by competing risks, they postulate the circadian rhythm as the source of the additional correlation. The combination of stochastic competition with copulas is original and interesting. Methods and statistical background are well described in the supplementary information.

Response: We thank the reviewer for the concise summary of our work and for the positive comments.

While the emergence and regulation of a circadian clock is an appealing hypothesis, the presented evidence that this mechanism is responsible for the observed correlations is from my perspective not convincing. Thus, the claim that 'circadian-controlled cell fate (is) inferred from single cell lineages' in title seems too bold and not justified given the data and results. Moreover, the data used for the inference should be better described and made available.

Response: We thank the reviewer for this comment. We have now developed a number of new models of cell fate determination based on stochastic protein production and degradation (explained in detail in response to reviewer #2 above). We show that none of these physically realistic models with non-oscillating mechanisms can explain the “cousin-mother inequality” that we observe in our data.

In addition, we have now also tested our model with various time periods of the oscillator besides the 24 hour period we had originally worked with. We show that only a few periods that are approximately multiples of 12 hours (including 24 hours, the independently measured circadian period of the cell line we used) can explain our data.

We believe that these new results in response to the reviewer's suggestions further bolster our proposal that an oscillatory mechanism, in particular the circadian clock, plays a role in cell fate control. As we discuss in our manuscript, this would also be consistent with many reports in the literature of a direct coupling of the circadian proteins to those that control cell cycle progression and cell death. Therefore, our work represents the most plausible description of the underlying biology as of now, providing concrete predictions that can be tested in future experiments. We hope that this will provide inspiration and direction for future studies aimed at more conclusive tests of the circadian gating model. However, we do concur that our work cannot completely rule out the possible existence of more complicated scenarios that may explain the data equally well, and we have now stated this in our Discussion section. **To reflect the additions we have made based upon the reviewer's suggestions, we have now also re-written the abstract.**

*** Major concerns

**** The circadian clock hypothesis

- Clearly, additional dependencies between sisters is needed to explain the observed correlations. While it is interesting to link this correlations to the circadian clock that has been described to exist autonomously in the cell line used, the evidence that this mechanism is indeed necessary to explain the observed data is missing. Other, non oscillating mechanisms, like DNA states or inherited proteome abundance could be equally suited to explain the observed correlations and have to be ruled out, e.g. via systematic model comparison.

Response: We thank the reviewer for this great suggestion. We are sorry if the following point was not clear in our original manuscript – the observation that the cousin correlations are larger than the mother-daughter correlations (termed the “cousin-mother inequality”) is the crucial aspect of the data that requires invoking an oscillatory mechanism.

To demonstrate this, we have now systematically developed a set of increasingly complex models of fate determination based on inherited protein abundance. These models are physically realistic and mimic our current understanding of the way in which inherited protein levels (or the combination of their levels) may affect cell fate probabilities. We show that none of these models can reproduce the cousin-mother inequality that we observe in our dataset. **These new models are explained in detail in response to Reviewer #2 above, and described in the main text, page 7, in SI section 6, and in two new figures, Fig. 4 and Fig. S6.**

- Experimental evidence is needed to prove the circadian clock hypothesis. E.g. via disturbing the circadian clock and re-evaluating the correlations with predictions, or via syncing the cells and directly observing changes in correlations and IMT over time.

Response: Please see our response to Reviewer 1 comment #5. In short, after discussion with our editor, we determined that the timescale of this revision and effort involved in conclusively

demonstrating this experimentally are outside the scope of this paper. This work will be the topic of future investigations.

- The authors assume a period of 24h for the circadian clock based on literature. Can the authors infer the period from the data? Can they evaluate if the period might change after drug addiction?

Response: We thank the reviewer for this great suggestion. We have now re-run our simulations for the pre-cisplatin scenario with varying periods for the oscillations. For every period, we chose the tunable parameters of our model such that the sister correlations and IMT distribution match the experimental observations. Interestingly, we find that *only* certain multiples of approximately 12 hour time-periods (12, 24, 48 hours; not 36) can reproduce the experimentally observed correlation structure. For all other periods that we tested (for example 3.5, 6, 14 and 18.5 hours), either one of two problems arose in explaining the data: (1) the mother-daughter correlations became strongly positive or (2) at very small time periods (around 4 hours), the cousin correlations became almost zero. These results are shown in the new **Fig. 6** and new **Fig. S9**. An intuitive explanation for the above observations is shown in **Fig. 6 a,c**. The mother-daughter correlation is set by the interplay between the average cell cycle length and the period of oscillations of the clock that gates the cell cycle. The HCT116 cell line has an approximately 16 hour average cell division time. For a period of about 18 hours, it is expected that the mother-daughter correlation will be positive (**Fig. 6 a,b**). For a period of about 3.5 hours representing very rapid oscillations, the heterogeneity in the cell division times will lead to randomly different phases at cell birth leading to negligible cousin correlation (**Fig. 6 c,d**). As we mentioned before, we could not rule out some of the other multiples of 12 as potential time-periods of the oscillations. This is likely due to the 95% confidence intervals on the correlations being fairly large in our data. **These results are now discussed in the main text, page 8, along with two new figures, Fig. 6 and Fig. S9.**

- It is unclear if the authors assume a global circadian clock for the entire cell-population or if each cell has its own phase. Furthermore, how is the phase of the circadian clock determined? Eq. 20 in the Supplement has 0 phase, implying that cell born at $\pi/2$ in the clock have longest cell cycle. How is this choice motivated?

Response: We thank the reviewer for this comment on the circadian phase of cells and apologize for not having explained this aspect of our model more clearly in the submitted version of the manuscript. In our original model, we did assume a global circadian clock across cells of all lineages. There was one universal clock that maintained an absolute time T , and the circadian phase was computed as $\Phi = \frac{2\pi}{48} T$ (48 frames = 24 hours). This was done purely for the sake of simplicity of the model and for reducing computing time and memory requirements – we needed to keep track of the phases of each lineage separately over time. This assumption was not meant to be physically realistic, as we pointed out in the SI, but to capture the minimal requirement that sisters have similar circadian phases. Since different lineages are independent

in birth-death models such as the one we used, for the purposes of our study it did not matter that we assumed identical phases across lineages.

To demonstrate the validity of our original results on the lineage correlations (in particular the cousin-mother inequality) even in the absence of this assumption, we developed a modified model: we chose the circadian phase of the 30 starting cells randomly from a uniform distribution between $[0, 2\pi]$. As a result, the phases of cells across lineages are no longer synchronized at any later time (**Fig. S8 a**). We show that our original results for the lineage correlations still hold in this modified version of the model (**Fig. S8 b**), proving that as long as the sister cells have similar circadian phases, the experimentally observed correlation structures can be quantitatively reproduced by our model. Note that in this modified model, the cells *within* a lineage still have synchronized circadian clocks, but we showed in the SI that adding small amounts of randomness in the passing of phase from mother to daughter does not change our results. **We have now added a sentence on Page 8 of the main text and the details of these simulations to SI section 5 along with a new Fig. S8.**

Regarding the reviewer's subsequent question regarding the form of the coupling function that sets the longest cycle time at a circadian phase of $\pi/2$: we broadly based our model on earlier findings in cyanobacteria, where the speed of the cell cycle was found to depend on the circadian phase but not on the cell cycle phase (Yang et al, Science 2010). The choice of longest cell cycle time at circadian phase = $\pi/2$ is a simplifying assumption we made in the absence of any current estimates of the coupling function for this particular cell type. Indeed, inferring the precise mathematical form of the underlying coupling function that connects the circadian phase to cell division speed is a challenging open problem that we will work to elucidate in future studies.

**** Results

- Using copulas and EMG, the "hidden" distributions of IMT are inferred from sister-pairs, taking into account correlations between related cells. In line 215-226, you claim that a simple simulation model (supplied with the inferred IMT distribution) cannot produce sister/sister correlations as seen in the data. This seems trivial, since you simulate sisters independently (any correlation would come from the competing risks). You used copulas to account for correlations going beyond the ones coming purely from competing risks. Hence the above would only work if the copulas would model independent random variables (i.e. not be needed in the first place). It is still an illustrative example, but you should state that this "finding" is expected. You could already conclude "that the origin of correlations [...] cannot be ascribed to the stochastic competition [...]" by looking at the copulas that were inferred.

Response: We apologize for the lack of clarity when discussing these results. For the post-cisplatin competing risks scenario, we did not jointly infer the various correlations along with the EMG parameters for division and death using our MCMC procedure. Jointly inferring the correlations would have required inferring at least four additional parameters (sister

correlations for division and death, and cousin correlations for division and death) beyond the 7 parameters we were already inferring (please see the SI for more details). The added parameters naturally complicate the inference procedure and we would have needed to develop more sophisticated MCMC algorithms to perform this task, as opposed to the Metropolis Hastings algorithm we used. Instead, we computed the Pearson correlations directly from the data and used them in our copula and competing risks framework to accurately infer the underlying IMT and AT distributions. We have now performed additional calculations to show that this procedure by itself (i.e. accounting for correlations) is expected to lead to a significantly better inference of the underlying IMT and AT distributions (please see our response to point 2 of Reviewer #1 and section 5 of the SI for more regarding this issue). This procedure therefore assumes the existence of sister and cousin correlations, without saying anything about the origin of such correlations. This is why we ran completely independent simulations -- to investigate whether the phenomenon of competing risks is sufficient to generate the observed correlations among the highly biased IMT and AT. We were not expecting competing risks to be able to generate the high correlations observed, but still performed these simulations for completeness. **We have now the material on page 6 of the main text to clarify this point.**

- Line 200-207: Simulating data from a model using the inferred parameters should not be called “validation” of the model or “proving the validity of our inferences”. It is a good sanity check (i.e. the model can recapitulate simulated data), but does not validate the model.

Response: We agree with the reviewer and have now **changed the material on page 6 of the main text to read “To check that these surprising results are plausible....” And “....providing an important sanity check of our inferences.”**

- Can the EMG-copula distribution fit the sister-IMT distribution adequately? It would be instructive to not only show the univariate fits as done throughout Fig. 3, but also the bivariate data/copula fit.

Response: We thank the reviewer for the suggestion, and have now **added a figure in the SI as well as a few lines in the main text (page 5) mentioning the bivariate density.** We used the inferred univariate EMG margins for the IMT in the copula formulation to generate simulated data and plotted the contours against the contours generated from the experimental data. As can be seen from **Fig. S4**, the EMG-copula approach provides a good description of the data.

****Presentation

- Neither in the results, nor in the Methods of the main text of the manuscript the model and inference procedure is described properly. Since this is done extremely nicely in the supplement, I would suggest to move some of the introduction and explanation to the main text.

Response: Thank you for this suggestion. **We have now added some details of the modeling (including two of the main equations) to the main text, pages 5 and 9.** We hope that the basic idea behind the modeling has been clarified with these additions. The full details remain unchanged in the **SI**.

- Material and Methods lack paragraphs on how cells were tracked and quantified over time. How do tracking errors affect the results? How many cells have been tracked, and how long? Where is the tracking data available? A movie showing cells and trackings would be instructive.
- The authors should show all used trees a supplementary figure.

Response: Thank you for this comment. **We have now added more details to the materials and methods section about our cell tracking.** In short, cell tracking was done with custom Matlab scripts that allow manual tracking using both a nuclear marker (H2B-ECFP) and phase contrast images to prevent tracking errors. Cell identities that were ambiguous were discarded to ensure the reliability of lineages. In total 394 cells were tracked from two biological replicates. The majority of these (85% or 337 cells) were tracked until they died or for the entire 5 day experiment. **These lineages are shown in a new supplemental figure, Fig. S14.** To increase the number of cells we included additional cells that we could not track at the beginning of the experiment but began tracking at a later timepoint until the cell died or the experiment ended. These lineages are now shown in Fig. S15. **All lineage data is now included in the supplemental file "SupplementalSingleCellData.xls". Following the reviewer's suggestion, we also added a time-lapse movie showing cells and tracking of 3 lineages.**

*** Minor issues

**** Presentation

- Please convert frames to hours and pixel to micron to make figures more informative

Response: We have now converted all figures axes from pixels to microns and from frames to hours. Please note that due to this rescaling of the time by a factor of 1/2, the Y-axes of all figures that display probability densities were rescaled by a factor of 2.

- The authors call their model 'mechanistic' at times. From my perspective, it is a purely statistical model.

Response: The copula combined with competing risks is indeed a purely statistical model aimed at better inference of the underlying IMT and AT distributions. However, the rest of the modeling, which involves the cell birth-death simulations coupled with the circadian gating model, is in our opinion a mechanistic model. This is mainly due to the fact that the circadian gating model is a generative model that can explain the emergence of the full correlation structure observed in the data based on underlying biological mechanisms. We have carefully checked the manuscript to clarify which models we refer to as mechanistic.

- Page 6: 'Novel unified theory' is rather a hypothesis I would say.

Response: Although we agree that we cannot conclusively pin down the circadian clock as the sole candidate for generating the observed correlation structures, we believe that our work represents much more than simply a 'hypothesis'. We indeed started out with the *hypothesis* that a coupling of the circadian clock to cell division and death could explain our data – but went further and created a computational model based on hazard functions that quantitatively reproduces all aspects of our data and generates experimentally testable predictions. In responding to the reviewer's comments, we have now shown that the underlying clock in our model is required to have an oscillation period of about 24 hours (or some other multiples of 12 hours) in order to explain our data. Other oscillation periods cannot recapitulate the data, strongly suggesting that the circadian clock indeed plays a role in cell fate control and generation of the correlations. Finally, again in response to the reviewer's suggestion, we have shown that models of stochastic protein production and degradation that best represent our current understanding of how cell division and death are controlled cannot explain the cousin-mother inequality we observe. Taken together, our combined experimental and theoretical approach provides much more than just a hypothesis of how cell fate is controlled. We therefore respectfully like to retain the phrase "unified theory".

- Fig. 4 and 5 could be merged.

Response: Thank you for the suggestion. Figure 4 represents the pre-cisplatin scenario while Figure 5 represents the post-cisplatin data analysis. We therefore believe that it will be easier for the reader to present this data and analyses separately.

- Fig 3abc would profit from the same data binning

Response: We thank the reviewer for this suggestion. Figs 3 a-c have dissimilar ranges on the x axis, and since we have limited data, we were not able to use the same size bins for all three figures. For example, if the same bin width used in 3a is used in 3c, then 3c becomes extremely noisy with many regions spuriously showing zero density.

- Fig 3bf and 3cg show the same fit, no? Please make comparisons feasibly by using the same ranges.

Response: The red curves in Fig 3b,f are identical and the green curves in 3c,g are identical. The histograms are not the same though – they represent the experimental data in 3 b,c and simulation data in 3f,g. **We have now explained this more carefully in the figure legend and have updated Fig. 3a and b to show the same ranges to enable direct comparisons.**

- Why are mother-daughter relationships not evaluated with the gated model and shown in Fig. 5?

Response: We did not have any data available for two full divisions representing the mother and daughter for cases in which the mother was born after cisplatin treatment. We therefore were not able to compute the mother-daughter correlations in Figure 5, which represents data purely in the post-cisplatin scenario.

- Fig 4c: Caption says 'cannot recapitulate the high correlations'. Would be instructive to show them in the figure as well.

Response: Thank you for the suggestion. Since the inset in this figure (**Fig. 5** in the revised version) takes up a lot of space, we did not show the 95% confidence intervals of the lineage correlations as measured from the experimental data. **We have now added a sentence in the figure caption directing the reader to panel (f) for the data.**

- Fig 4c: Can the authors explain the large variability of cousin-cousin correlations vs. μ ?

Response: The variability in the cousin correlations essentially arises from the fact that these are stochastic simulations. Since the number of cousin pairs observed in our experiment was not very large (46 pairs), and we simulated roughly the same number of pairs, there is some amount of variability between different simulation runs. However, only when μ_{die} becomes comparable to μ_{div} or lower (i.e., death occurs faster than division) are the IMT correlations affected, as we demonstrate later in the manuscript.

- Fig 4c inset: What is the observed number of dead cells?

Response: We did not track cell death prior to cisplatin treatment in this study. However, we have done this in a previous study, and found that over a period of 72 hours, approximately 8% of HCT116 cells die (Fig. S1D, Paek et al, Cell 2016). Therefore in Fig. 5c (previously Fig. 4c) we varied the death rate such that we observed up to 8% cell death approximately (please note that here we added cisplatin after 48 hours). **We have now added a line in page 8 of the main text discussing this issue.**

- Methods say: 'Cell cycle stage was determined by visual inspection of Cer-hGem levels'. Where is that used?

Response: This analysis is shown in the supplement. We have now moved this description to the supplemental experimental procedures to avoid confusion.

- What are the fluorescent readouts (Venus and mCherry)? Tracking was performed on brightfield I suppose?

Response: We apologize for not being clear on this. For experiments described in the main paper, the fluorescent readouts are Venus for p53 levels, while H2B-ECFP was used as a nuclear marker. Tracking was performed with a combination of the brightfield and the ECFP channel to ensure accuracy. This is now outlined in more detail in the materials and methods. For the

supplemental experiments on the cell cycle, we used Venus for p53 levels and Cerulean for tracking the hGem reporter. Here cells were tracked using the phase channel. **This has now been covered in more detail in the supplemental experimental procedures.**

- Scale bars are missing in Fig 1a.

Response: Apologies for omitting this; we have now added scale bars to Fig 1a.

- Colors in Fig 1cde are unfortunate. Since this all corresponds to cells before treatment, I suggest to use violet throughout.

Response: The colors in Fig. 1 c-e are consistently the same for mother-daughter, sister-sister and cousin-cousin correlations across all figures in the manuscript. We feel that differentiating the lineage correlations with different colors is helpful. **We have now clarified the color code in the figure legend.**

**** Results

- Page 3: Pearson correlations need error bars, either from replicates or at least from bootstrapping. I specifically doubt the -0.03 for mother-daughter correlations, which seems to be strongly influenced by two extreme values in Fig. 1c

Response: Thank you for this comment. We provided the 95% confidence intervals in the main text as well as in figures 4 and 5 (dashed lines). **We have now added these confidence intervals to the caption of Figure 1.**

- Page 4: You can only assume 50% same fates in sisters if the fates are equally frequent, no?

Response: For the experiment, 62.5% of cells died in response to cisplatin treatment so ~53% of cells should share the same fate if the response of each cell was an independent event. This value was calculated with the probability of both cells dying is ~39% ($.625 * .625$) and the probability of both cells surviving is ~14% ($.375 * .375$). **We have now included these calculations in the supplemental section “Cell fate is correlated in related cells”.** In this section, we also calculate that only 52% of unrelated cells in our experiment share the same fate, in line with the estimates above.

REVIEWERS' COMMENTS:

Reviewer #1 (Remarks to the Author):

The authors have done a thorough work addressing comments of all reviewers. The manuscript has been improved and shows that the circadian clock is a very plausible hypothesis which should trigger further work. Only one comment remains to be addressed about the influence of the circadian model of cell cycle distributions: the work that does that is not the one that is quoted (Mosheiff, bioarchive) but the final version of that work published in PRX 2018.

Reviewer #2 (Remarks to the Author):

The revised manuscript has been extended by additionally testing reasonable models for cell cycle control by inheritance of protein levels, in order to check whether such models could also reproduce the experimental data. Furthermore, the authors have now also measured cousin-cousin and mother-cousin correlations in IMTs and compare their models with this data. They show that such protein-level-inheritance models are not consistent with the cousin-cousin and mother-cousin correlations. It is thus most likely that indeed an internal clock mechanism, such as the Circadian clock, are controlling the IMTs. The results are reasonable. However, I was a bit confused about the claims that the model matches the data in Figs. 6d and 7b,d where it looks like some model predictions lie outside of the confidence intervals, or very close to its boundary. While this is indeed possible by chance, it would be good if the authors could quantify this statistically, to exclude significant deviations.

If the authors can resolve any issues concerning the match of the Circadian model in Figs. 6 and 7, and some minor issues remarked below, I would be satisfied and recommend publication in Nature Communications.

Some comments:

1. It would be helpful to have units for the protein production and degradation rates in the protein-inheritance model (in the SI), in order to compare the degradation time scale (the time scale of "memory") with the cell cycle length.

2. In Fig. 4, it would be helpful to also see the experimental data to have a direct comparison between model output and data.

Reviewer #3 (Remarks to the Author):

The authors have satisfactorily addressed all issues and deliver a comprehensive, thoroughly revised manuscript. They might consider citing our recently published work on inference of molecular mechanisms from lineage correlations (although not related to circadian rhythms): <https://www.nature.com/articles/s41467-018-05037-3>

REVIEWERS' COMMENTS:

Reviewer #1 (Remarks to the Author):

The authors have done a thorough work addressing comments of all reviewers. The manuscript has been improved and shows that the circadian clock is a very plausible hypothesis which should trigger further work. Only one comment remains to be addressed about the influence of the circadian model of cell cycle distributions: the work that does that is not the one that is quoted (Mosheiff, bioarchive) but the final version of that work published in PRX 2018.

Response: Thank you. We have now updated the reference to the PRX version.

Reviewer #2 (Remarks to the Author):

The revised manuscript has been extended by additionally testing reasonable models for cell cycle control by inheritance of protein levels, in order to check whether such models could also reproduce the experimental data. Furthermore, the authors have now also measured cousin-cousin and mother-cousin correlations in IMTs and compare their models with this data. They show that such protein-level-inheritance models are not consistent with the cousin-cousin and mother-cousin correlations. It is thus most likely that indeed an internal clock mechanism, such as the Circadian clock, are controlling the IMTs. The results are reasonable. However, I was a bit confused about the claims that the model matches the data in Figs. 6d and 7b,d where it looks like some model predictions lie outside of the confidence intervals, or very close to its boundary. While this is indeed possible by chance, it would be good if the authors could quantify this statistically, to exclude significant deviations.

Response: Thank you for this comment. Since our simulations generating single cell lineages are based on stochastic dynamics, there will naturally be some amount of scatter in the data shown in Figs 6d and 7b,d. To demonstrate that the relatively limited statistics we generated in these figures (25 simulation runs in each boxplot) did not result in chance large deviations generating misleading results, we have now rerun the analyses with 50, 100 and 150 simulation runs each. In each case we obtained results that are approximately the same as in our current Figs 6d and 7b,d. For Fig 6d, we found that only ~ 7% of the simulation runs result in correlations that lie within the experimentally obtained 95% confidence intervals, suggesting that the model with an oscillation time period of 3.5 hours does *not* explain the data. On the other hand, as in our current Fig. 7b,d representing the results of a model incorporating circadian gating, we found that approximately 40% and 75% of the simulated IMT correlations for sisters and cousins, respectively, and approximately 78% and 55% of the simulated AT correlations for sisters and cousins, respectively, fall within the 95% confidence intervals of the data. These observations demonstrate that the results we reported in our original Figs 6d and 7b,d are not outcomes of chance fluctuations. In addition, we had shown that a 'null model' with only stochastic competition of fates and no circadian gating cannot explain the observed correlations after cisplatin treatment (Figs. 7a,c) – in this case, the median correlations lie well outside the confidence intervals of the data. **We have now added a line to the caption of Fig 7 explaining this point.**

If the authors can resolve any issues concerning the match of the Circadian model in Figs. 6 and 7, and some minor issues remarked below, I would be satisfied and recommend publication in Nature Communications.

Some comments:

1. It would be helpful to have units for the protein production and degradation rates in the protein-inheritance model (in the SI), in order to compare the degradation time scale (the time scale of "memory") with the cell cycle length.

Response: Thank you for this suggestion. **We have now added the units of the production and degradation rates to the SI.**

2. In Fig. 4, it would be helpful to also see the experimental data to have a direct comparison between model output and data.

Response: Thank you for this suggestion. **We have now added the 95% confidence intervals of the experimental data to Fig. 4.**

Reviewer #3 (Remarks to the Author):

The authors have satisfactorily addressed all issues and deliver a comprehensive, thoroughly revised manuscript. They might consider citing our recently published work on inference of molecular mechanisms from lineage correlations (although not related to circadian rhythms): <https://www.nature.com/articles/s41467-018-05037-3>

Response: We thank the reviewer for pointing out the very interesting reference and **have now incorporated it into our manuscript.**